# Integration of single-cell transcriptomes and chromatin landscapes reveals regulatory programs driving pharyngeal organ development

Margaret E. Magaletta[1,2,6], Macrina Lobo[1,2,6], Eric M. Kernfeld [1,2,6], Hananeh Aliee[3], Jack D. Huey[1,2], Teagan J. Parsons[1,2], Fabian J. Theis [3,4,5] & René Maehr [1,2✉]

Maldevelopment of the pharyngeal endoderm, an embryonic tissue critical for patterning of the pharyngeal region and ensuing organogenesis, ultimately contributes to several classes of human developmental syndromes and disorders. Such syndromes are characterized by a spectrum of phenotypes that currently cannot be fully explained by known mutations or genetic variants due to gaps in characterization of critical drivers of normal and dysfunctional development. Despite the disease-relevance of pharyngeal endoderm, we still lack a comprehensive and integrative view of the molecular basis and gene regulatory networks driving pharyngeal endoderm development. To close this gap, we apply transcriptomic and chromatin accessibility single-cell sequencing technologies to generate a multi-omic developmental resource spanning pharyngeal endoderm patterning to the emergence of organ-specific epithelia in the developing mouse embryo. We identify cell-type specific gene regulation, distill GRN models that define developing organ domains, and characterize the role of an immunodeficiency-associated forkhead box transcription factor.

[1] Program in Molecular Medicine, University of Massachusetts Medical School, Worcester, MA, USA. [2] Diabetes Center of Excellence, University of Massachusetts Medical School, Worcester, MA, USA. [3] Institute of Computational Biology, Helmholtz Zentrum München, Munich, Germany. [4] Department of Mathematics, Technische Universität München, Munich, Germany. [5] School of Life Sciences Weihenstephan, Technische Universität München, Freising, Germany. [6] These authors contributed equally: Margaret E. Magaletta, Macrina Lobo, Eric M. Kernfeld. ✉email: rene.maehr@umassmed.edu

Development of the pharyngeal endoderm, which produces specialized epithelia of various organs including the thymus, parathyroid, thyroid, ultimobranchial body, the middle ear, and palatine tonsils[1–3], is characterized by dramatic changes in morphology and cell state across embryonic days (E) 9.5 to 12.5 of mouse gestation. At E9.5, pharyngeal endoderm comprises four bilateral out-pockets organized from anterior to posterior, termed pharyngeal pouches I–IV. Each pouch forms the epithelial structure of a unique organ that separates from the pharynx by E12.5. Importantly, disorders associated with defects in pharyngeal endoderm development are characterized by a spectrum of phenotypes ranging from craniofacial dysmorphia, heart deformities, and cognitive deficiencies to severe immune disorders and disrupted endocrine processes. Such symptoms arise in part due to dysfunctions of the pharyngeal endoderm in organization of the pharyngeal region, and cell autonomous regulation of patterning and organogenesis[4–9]. Certain mutations or single-nucleotide polymorphisms (SNPs) in transcription factors (TFs) or *cis*-regulatory elements (CREs)[10–13] are known to cause pharyngeal endoderm-associated syndromes, but do not fully account for all incidences of such syndromes. Moreover, large variations in phenotypic penetrance suggest contribution of unknown genetic modifiers to observed phenotypes[14,15]. Given the developmental origin of these syndromes, a comprehensive understanding of pharyngeal endoderm development could lead to specific hypotheses about molecular mechanisms driving syndrome emergence and phenotypic penetrance variation; however, a detailed integrative molecular characterization of pharyngeal endoderm development does not yet exist.

Single-cell genomics technologies provide means to dissect the molecular basis and gene regulatory networks (GRNs) driving cell development and heterogeneity, and thus could provide a comprehensive approach to reveal potential coding and non-coding factors that could impact syndromes and phenotype penetrance. While others have reported single-cell transcriptomic profiles covering early endoderm development or whole embryo development[16,17], these studies fall short of capturing the time period of pharyngeal endoderm patterning and subsequent organ formation or lack sufficient cell numbers. Moreover, most of the aforementioned studies focus exclusively on transcriptomic analyses, neglecting chromatin aspects of this developmental time period, and only recent reports have begun to capture single-cell chromatin accessibility maps on whole mouse embryos including endodermal cells[18], albeit not including pharyngeal endoderm development and pharyngeal organogenesis.

To address the knowledge gap, we present a comprehensive single-cell transcriptomic and chromatin accessibility resource on mammalian pharyngeal endoderm development spanning the time period of pharyngeal patterning to the emergence of organ-specific epithelia. This resource provides a description of tissue-specific coding and non-coding elements, some of which we characterize spatially. We reveal GRNs that distinguish developing organ domains and predict the effects of perturbations on pharyngeal endoderm development. Finally, we substantiate GRN model predictions by characterizing molecular and cellular impact of a perturbation targeting *Foxn1*, which encodes a forkhead box DNA binding TF, resulting in developmental defects that underlie primary immunodeficiency.

## Results
### Single-cell RNA-seq reveals cellular heterogeneity and transcriptomic dynamics of pharyngeal endoderm development. To investigate the molecular processes underlying development of pharyngeal endoderm, we generated a single-cell transcriptomic catalog of mouse pharyngeal endoderm between embryonic days

(E)9.5 and E12.5 which covers the transition from pharyngeal endoderm to pharyngeal organ primordia[2] (Fig. 1a). To specifically isolate pharyngeal endoderm, we sorted Epcam/Pax9VE-NUS cells as previously described by us[19]. After initial data processing and quality control of the single-cell RNA (scRNA) profiles (see "Methods"; Supplementary Figs. 1 and 2), the resulting dataset yielded 54,044 single-cell transcriptomes (13,345 from E9.5, 13,120 from E10.5, 16,493 from E11.5, 11,086 from E12.5), with a median of 3072 unique genes detected per cell.

Dimensional reduction and unsupervised clustering identified 28 distinct clusters, with cells of earlier timepoints, hence less mature pharyngeal endoderm, grouped in the center of the UMAP embedding and cells of later timepoints extending to the periphery, representing organ specification and maturation events (Fig. 1b, Supplementary Fig. 3, Supplementary Table 1). Differential gene expression analysis revealed signatures of various organ domains in the peripheral clusters, including the eustachian tube, thyroid, thymus (cortex and medulla), parathyroid, ultimobranchial body (UBB), oropharynx, and esophagus (Fig. 1c–e, Supplementary Fig. 3, Supplementary Dataset 1).

Cluster-specific molecular signatures related to the thyroid (*Nkx2.1, Hhex, Pax8*[20]), early respiratory lineages (*Nkx2.1*[21], *Irx1/2*[22]), and first pouch derivative Eustachian tube (*Fgf8 & Edn1*[23], *Eya1*[24]) is detected as early as E9.5 (Supplementary Fig. 3). UBB cells become distinct at E10.5 (*Nkx2.1, Hhex, Calca*[25]), supporting the pharyngeal endoderm origin of UBB[26] (Supplementary Fig. 3). Interestingly, the expression pattern of *Ripply3*, a known marker of the fourth pouch[27], points to a UBB progenitor-containing cluster at E9.5.

The second pouch cluster comprising mainly E10.5 and E11.5 cells expresses genes associated with salivary glandular epithelium (*Irf6*[28], *Trp63, Sox9, Vim*[29]). Within the same cluster, E12.5 cells express certain markers of myoepithelial cells (*Acta2, Tagln*[29], *Myl9*[30]), which are documented in various epithelial glands[31]. This cluster also expresses pharyngeal pouch markers *Pax1* and *Fgf8*[32,33], but lacks expression of *Hoxa3*, a known marker of the caudal third and fourth pouches[34], suggesting an anterior position (Supplementary Fig. 3). Notably, morphological similarities between the salivary gland and palatine tonsils[35], the known derivative of the second pharyngeal pouch in most mammals, support the hypothesis about an evolutionary relationship between the two tissues[36].

In agreement with previous reports[37,38], parathyroid (*Gcm2*) and thymus (*Foxn1*) appear by E10.5 and E11.5, respectively (Supplementary Fig. 3). At E12.5, thymic epithelial cells (TECs) could be further separated into two clusters, with cluster 25 expressing *Cldn3* and *Cldn4*, markers associated with the emergence of specialized medullary (m)TECs[39], and cluster 4 enriched in expression of *Psmb11*, a transcript associated with cortical (c)TECs and progenitor TECs[40] (Fig. 1e, Supplementary Fig. 3). Also at E12.5, we observe clusters that likely correspond to oropharynx and esophageal epithelium (*Sox2, Trp63, Krt15*[41]).

In addition, unbiased transcriptomic profiling exposed cell-type-specific gene expression patterns (Fig. 1e, Supplementary Fig. 3, and Supplementary Dataset 1). To verify these findings, we implemented RNAscope at E12.5. Using *Foxn1, Gcm2,* and *Calca* probes to demarcate the thymus, parathyroid, and UBB, respectively, we co-stained with predicted transcripts enriched in the thymus (*Gas6, Grhl3*), parathyroid (*Sparcl1, Ibsp, Flrt2*), and UBB (*Meox1*) (Fig. 1f–h, Supplementary Fig. 3). While *Gas6* expression overlaps with *Foxn1* throughout the thymus organ domain, we find that *Grhl3* expression is largely restricted to the thymic medulla, which together with the single-cell transcriptomic analysis suggests mTEC specificity (Fig. 1f). In the parathyroid, we observe high expression levels of *Sparcl1* (Fig. 1g), a gene frequently downregulated in epithelial cancers, including

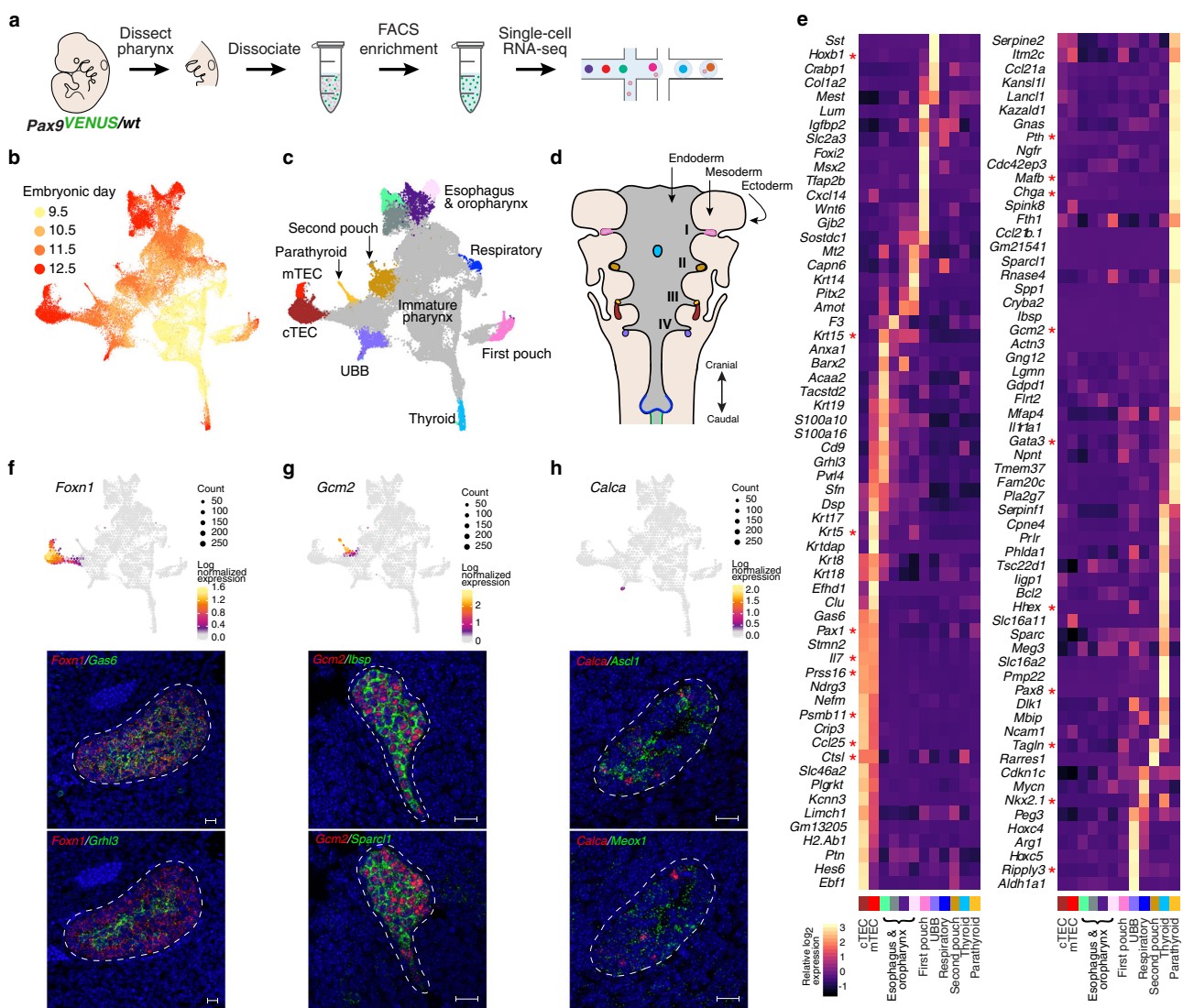

**Fig. 1 Single-cell transcriptomic map of developing mouse pharyngeal endoderm from pouch formation to early organogenesis. a** Experimental workflow schematic. $Pax9^{VENUS}$ embryos were harvested at E9.5 ($n = 2$), E10.5 ($n = 3$), E11.5 ($n = 3$), and E12.5 ($n = 2$), and the pharyngeal region was dissected. Tissue samples were dissociated into a single-cell solution and then pharyngeal endoderm cells were FACS purified based on co-expression of VENUS and Epcam. Single-cell transcriptomes were captured and barcoded using Chromium Single-cell 3′ Reagents. **b**, **c** UMAP visualization of pharyngeal endoderm transcriptomic time-course dataset ($n = 54,044$ cells) colored by embryonic day (**b**) and terminal Louvain clusters corresponding to anatomical structures (**c**). Each dot represents a single-cell in the global transcriptomic space. mTEC medullary thymic epithelial cells, cTEC cortical thymic epithelial cells, UBB ultimobranchial body. **d** Diagram of pharyngeal endoderm with pharyngeal pouches I–IV indicated. Organ colors correspond to cluster colors in (**c**). **e** Heatmap displaying differentially expressed marker genes by terminal clusters. Row-standardized heatmap of data-driven terminal cluster marker genes. Genes displayed represent a log2-fold-change cutoff of >1.5 and a p-value of <0.01. **f–h** UMAP visualization of expression of select tissue-specific marker genes with corresponding RNAscope data. Cells are binned by UMAP coordinate with size proportional to the number of cells and color signifying average log2-normalized expression. Data-driven and known transcripts of the thymus (data-driven: Grhl3, Gas6; known: Foxn1; $n = 3$) (**f**), parathyroid (data-driven: Ibsp, Sparcl1; known: Gcm2; $n = 3$) (**g**), and ultimobranchial body (data-driven: Ascl1, Meox1; known: Calca; $n = 2$) (**h**) lineages were visualized using RNAscope in situ hybridization. Scale bars represent 20 μm.

parathyroid cancer, leading to increased proliferation and cell cycle progression[42]; therefore, our results indicate this gene could be performing an anti-growth or anti-proliferative function during early parathyroid development.

**Single-cell chromatin accessibility profiling uncovers chromatin dynamics accompanying transcriptomic changes during pharyngeal organ specification.** Since single-cell transcriptomic analyses highlighted organ domain-specific expression signatures at E11.5 and E12.5, we aimed to further elucidate the regulatory landscape of pharyngeal endoderm development during this time period by measuring single-cell chromatin accessibility (Fig. 2a).

After data preprocessing and cleaning (see "Methods", Supplementary Fig. 4, Supplementary Fig. 5), we retained 10,890 cells (4323 from E11.5 and 6567 from E12.5) with 48.9% of reads in peaks (median) and 20.9% in promoters (median). These data were re-processed by dimensional reduction and unsupervised clustering by chromatin accessibility profiles, yielding 22 clusters over two developmental timepoints as visualized on the two-dimensional UMAP embedding (Fig. 2b, c).

To assign cellular identities, we integrated predicted gene expression scores, which were calculated based on gene body (upto 5 kb upstream of the transcription start site (TSS)) and putative distal regulatory element (100 kb on either side of the

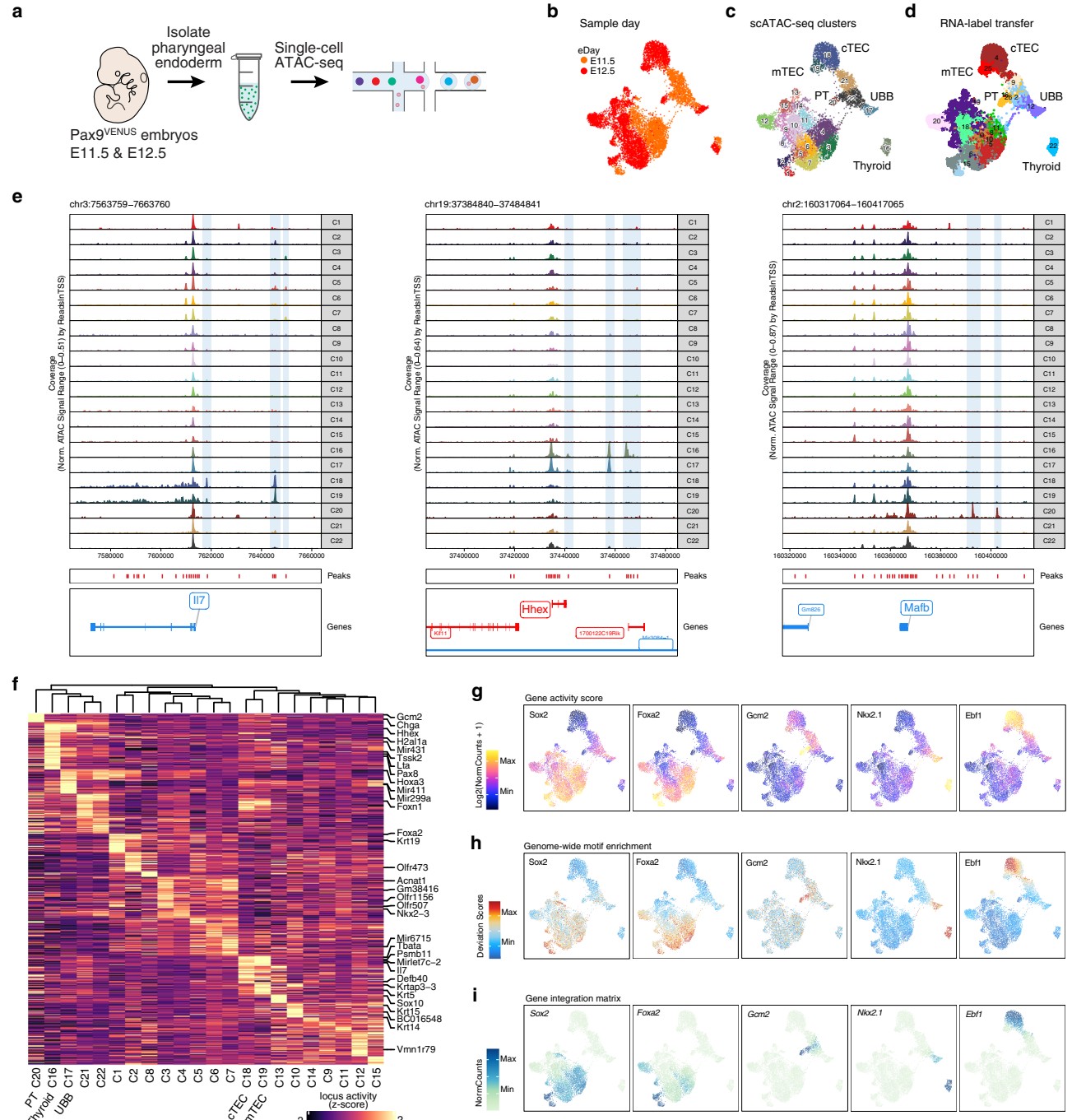

**Fig. 2 Single-cell chromatin map of developing mouse pharyngeal endoderm during early organogenesis. a** Experimental workflow schematic. Single cells were prepared as described in Fig. 1a from E11.5 ($n = 2$) and E12.5 ($n = 2$) Pax9$^{VENUS}$ embryos. Single-cell chromatin landscapes were captured and barcoded using Chromium Single-cell ATAC Reagents. **b–d** UMAP visualization of pharyngeal endoderm early organogenesis chromatin accessibility dataset ($n = 10,890$ cells) colored by embryonic day (**b**), Louvain clusters (**c**), and scRNA label transfer (see Supplementary Fig. 3 for cluster color labels) (**d**). Each dot represents one cell in the global chromatin accessibility space. cTEC, cortical thymic epithelial cells; mTEC, medullary thymic epithelial cells; PT, parathyroid; UBB, ultimobranchial body. **e** Normalized genome browser tracks of three tissue-specific loci (*Il7*, *Hhex*, and *Mafb*) highlighting annotated peaks. Each track displays pseudobulk ATAC data aggregated and colored by Louvain cluster from (**c**). **f** Heatmap of differential ($p$-value ≤ 0.1, log2-fold change ≥ 0.5) cluster-specific predicted gene expression (ArchR gene score) of the top 50 genes per cluster, row z-scored with clusters ordered by unsupervised hierarchical clustering. **g–i** Tissue-specific TF activity through gene activity score (**g**), genome-wide motif enrichment (**h**) and, inferred RNA expression based on the gene integration matrix (**i**).

gene body, while excluding promoter regions) accessibilities, with the scRNA gene expression data. This identified pharyngeal derivatives including the thyroid, TECs, parathyroid, UBB, oropharynx, and esophagus (Fig. 2d). Cluster-based pseudobulk ATAC-seq signal reveals locus-specific chromatin accessibility at

gene bodies linked to function and organogenesis of different pharyngeal endoderm derivatives (Fig. 2e). Importantly, putative cell-type-specific regulatory elements were captured as exemplified in the distal regions of *Il7*, *Hhex*, and *Mafb* genes, each of which performs critical functions in development and function of

the thymus, thyroid, and parathyroid, respectively[43–45] (Fig. 2e, shaded regions). In concordance, unbiased assessment of the differential gene scores (FDR ≤ 0.1, log2-fold change ≥ 0.5) by cluster revealed many known tissue-specific genes (Fig. 2f, Supplementary Dataset 2).

Last, we detected enrichment of known transcription factor (TF) binding motifs in chromatin accessibility peaks and compared UMAP visualizations of predicted gene expression scores with corresponding motif enrichment at the single-cell level, and matched scRNA gene expression based on the integration (Fig. 2g–i), revealing strong overlap between each of these data features in a cell-type-specific manner.

To uncover putative regulatory elements in the non-coding genome of each cell type, we annotated 270,210 peaks as promoter (2000 bp upstream to 100 bp downstream of a TSS, 9.3% of all peaks) or distal elements (peaks outside the gene body that are not promoters, 38.6% of all peaks). Parsing 64,518 differentially accessible peaks across clusters (FDR ≤ 0.01, log2-fold change ≥ 0.5) by promoter (3.8%) and distal (42.3%) highlighted greater diversity in distal peaks (Fig. 3a, Supplementary Dataset 3). Further, we determined peak specificity using the tissue specificity index[46], with smaller values indicating higher average peak signal and larger values indicating lower average peak signal across the clusters. We observed a higher cluster specificity of distal peaks, suggesting that cell identity in terms of the chromatin accessibility landscape is best ascribed to non-coding distal elements of the genome (Fig. 3b).

Within the differentially accessible set of distal peaks (FDR ≤ 0.1, log2-fold change ≥ 0.5), we identified cluster-specific enriched motifs and paired each with a candidate regulatory TF of the same family based on correlation between motif enrichment and scRNA expression (Fig. 3c). This approach identified several putative TF regulators of the pharyngeal endoderm. For example, Trp63 was enriched across many clusters, which is notable given its function in differentiation and/or proliferation of various epithelial stem cells[47,48]. TF motifs enriched in specific clusters include Pax8 and Nkx2.1 in the thyroid or Isl1 in the thymus and UBB. This approach also suggests Grhl3, a putative mTEC TF (Fig. 1), could be a regulator in both mTECs and the esophagus domain based on a high correlation with the Grhl1 motif.

Considering the relevance of the pharyngeal endoderm with respect to human developmental syndromes, we sought to further examine the distal peaks for evidence of conserved genetic programs between mouse and human that could contribute to unique cell identity across both species. To this end, we identified distal peaks conserved within the Euarchontoglires clade and differentially accessible by cluster (FDR ≤ 0.01, log2-fold change ≥ 0.5) (Fig. 3d). We then conducted GREAT analysis on each set of conserved cluster-specific distal peaks to uncover the biological pathways and phenotypes (Fig. 3e), and curated enriched terms to assess differential presence of ontologies. Interestingly, ontologies with key words such as pharyngeal pouch and pharyngeal/brachial abnormalities grouped together in the putative caudal pouch-containing clusters (clusters 21 and 22). This analysis also highlighted many terms related to development, function, and disorders of specific pharyngeal endoderm tissues. For example, the thymus cluster contains peaks near genes linked to immune system terms and Wnt signaling, which had previously been linked to thymus development[49], and the thyroid cluster contains peaks near genes linked to thyroid agenesis.

**Gene regulatory network inference highlights organ domain-specific regulatory nodes and edges.** To infer GRNs that could coordinate cell-type differentiation, we chose two approaches

(Fig. 4a). First, to uncover global patterns of regulation across the cell types in the developing pharyngeal endoderm we grouped cells with similar transcriptomic profiles into metacells and used these to build a GRN with GENIE3[50], which uses tree-based importance measures to infer causal TF-target networks. Unsupervised clustering on the network containing the top 200 regulators yielded 12 subnetworks (Fig. 4b, Supplementary Fig. 6, Supplementary Dataset 4), and we identified key TF-encoding genes of these modules including Pax8 (Thyroid), Foxn1, Pax1, and Six1 (Thymus), and Gcm2, Gata3, and Maf (Parathyroid) by ranking regulators according to their ability to explain coordinated variation within the pharyngeal endoderm data (Fig. 4c). Some of these subnetworks are highly cell-type specific, suggesting a capture of unique drivers of pharyngeal endoderm differentiation, while other subnetworks capture programs active across multiple cell types (Fig. 4d, e, Supplementary Fig. 6).

While the GENIE3 network successfully identified ubiquitous programs and certain cell-type-specific programs, the results could be difficult to interpret when one TF-encoding gene performs multiple roles in different cell types. For instance, Nkx2.1 regulates both UBB and thyroid development, and thus will have connections relevant to each cell type that could be indistinguishable. Therefore, to explicitly study cell-type-specific gene regulatory programs, the second approach we implemented was CellOracle[51], a network inference tool that capitalizes on both scRNA expression data and peak co-accessibility from single-cell ATAC-seq data to build cluster-specific GRNs and simulate network perturbations such as TF knockouts. A bivariate plot of betweenness centrality, a measure of gene importance within a network, and expression specificity yields genes known to regulate development of the thymus (Foxn1, Pax1), parathyroid (Gcm2, Mafb), UBB (Nkx2.1), and thyroid (Nkx2.1, Pax8, Foxe1), suggesting that regulators are best ranked by both specificity and regulatory influence (Fig. 4f, Supplementary Dataset 5). As expected, Nkx2.1 appears as a top regulator in both the UBB and thyroid CellOracle networks. Finally, this approach reveals several TF-encoding genes without previously described roles in each respective tissue: Nfia (thymus), Scx (parathyroid), Prdm1 (UBB), and Mecom (thyroid). Given that a recent publication demonstrates that Mecom gene products interact with Pax8 to drive gene expression programs regulating cell adhesion and extracellular matrix formation[52], our results could suggest a comparable interaction between Mecom and Pax8 to coordinate similar gene expression programs in the developing thyroid. Betweenness centrality of certain top regulators across all clusters revealed a distinctly cell-type-specific pattern, demonstrating the reliability of the CellOracle networks (Fig. 4g).

**Foxn1 deficiency causes thymus-specific transcriptional changes similar to a developmental delay.** To predict the importance of several organ-specific nodes from our GRNs in silico, we simulated genetic knockouts with the CellOracle GRN (Supplementary Fig. 7). One of the critical nodes identified in both the GENIE3 and CellOracle GRNs for TECs was Foxn1, a TF linked to thymus development and immunodeficiency[53]. Foxn1 knockout (KO) mice fail to develop a functional thymus[54], and studies suggest that the epithelial cells within the thymic rudiment arrest in a bipotent progenitor state[55]. Indeed, upon simulating a Foxn1 KO perturbation by propagating the downstream effect on the predicted direct and indirect targets using the inferred CellOracle GRN, the simulation proposed a developmental block of TEC differentiation (Fig. 5a). While evidence suggests that Foxn1 controls thymus differentiation rather than lineage commitment[56], questions remain about the molecular state of TECs in the Foxn1 KO. Earlier morphological characterizations of the Foxn1 KO

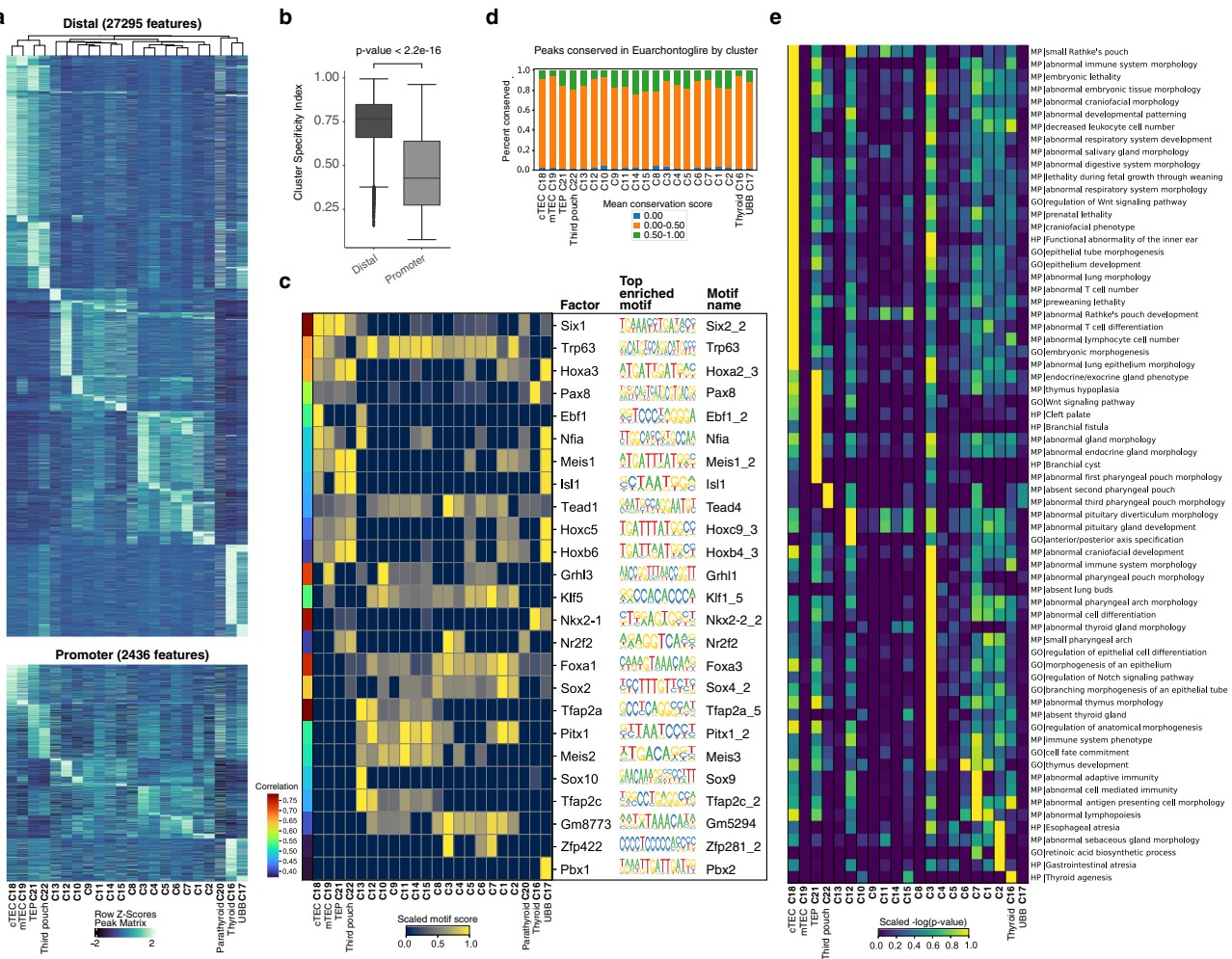

**Fig. 3 Distal peaks lend cell-type uniqueness and reveal underlying cell-type-specific pathways and phenotypes. a** Heatmap of differential cluster-specific peak accessibility (FDR ≤ 0.01, log2-fold change ≥ 0.5) parsed into distal ($n = 27{,}295$ peaks) and promoter ($n = 2436$ peaks) with clusters ordered by unsupervised hierarchical clustering on the distal peaks and row z-scored. TEP, thymic epithelial progenitor. **b** Boxplot showing cluster specificity of differentially accessible distal ($n = 27{,}295$) and promoter proximal ($n = 2436$) peaks measured by cluster-specificity index on scaled, log2-transformed cluster averaged peak accessibility. Center line denotes the median; box limits denote upper and lower quartiles; top and bottom whiskers denote regions upto 1.5 times the inter-quartile range above the upper and below the lower quartiles, respectively; tailing points denote outliers. *P*-value is from a two-sided Wilcoxon test with continuity correction and without adjustment. **c** Heatmap of top 10% of distal regulators per cluster determined by a motif score (average ChromVar motif z-score per cluster – minimum cluster average motif z-score) on a set of 55,840 differentially accessible distal peaks (FDR ≤ 0.1, log2-fold change ≥ 0.5) with color bar displaying correlation of the ChromVar motif z-scores with the integrated gene expression (correlation cutoff > 0.35). **d** Percent of distal differentially accessible peaks between clusters (FDR ≤ 0.01, log2-fold change ≥ 0.5) having an average phastCon conservation score within the Euarchontoglires clade = 0.00, >0.00 and ≤0.50, and >0.50 split by cluster. Cluster 20 has no peaks meeting the differential accessibility cutoff. **e** Heatmap of row normalized −log10 corrected *p*-values (Benjamini–Hochberg) from the binomial test on a curated list of GREAT terms enriched in distal, differentially accessible peaks (FDR ≤ 0.01, log2-fold change ≥ 0.5) with an average phastCon conservation score within the Euarchontoglires clade of >0.50.

thymus (a.k.a. nude thymus) revealed a cystic, alymphoid rudiment[57] with features similar to respiratory epithelium[58], while more recent work characterizing the nude thymus implicates *Foxn1* as an inhibitor of branching morphogenesis[59]. Nevertheless, the causative genetic defects supervening *Foxn1* deficiency are not yet fully understood, so we sought to characterize *Foxn1* KO during early organogenesis using our reference data as a framework.

To test the predictions of the *Foxn1* KO simulation created with CellOracle and thoroughly assess *Foxn1* deficiency in the early thymic epithelium, we implemented single-cell RNA-seq to characterize *Foxn1* KO and *Foxn1* heterozygous pharyngeal endoderm (Pax9$^{VENUS}$/Epcam positive) at E12.5. Notably, comparison of thymus resident CD4 and CD8 T cell populations

from wild-type, *Pax9$^{VENUS/wt}$* and *Pax9$^{VENUS/wt}$Foxn1$^{nu/wt}$* mice indicated that double heterozygous thymi retain normal function (Supplementary Fig. 8). After depletion of contaminating cells and empty droplets (Supplementary Fig. 9), 21,904 cells (9063 cells from *Foxn1* het and 12,841 cells from *Foxn1* KO) remained, with a median of 3969.5 unique genes detected per cell. Next, we projected the *Foxn1* KO and control cells onto the transcriptomic atlas and assessed cellular distribution (Supplementary Fig. 10). *Foxn1* KO cellular heterogeneity largely mirrored the E12.5 atlas samples regardless of genotype, except the *Foxn1* KO samples lack the most mature TECs (Fig. 5b, c). Notably, while the *Foxn1* heterozygous control sample appeared more mature than the *Foxn1* KO, the control included less mature TECs as compared to the atlas, a result that agrees with previous descriptions of

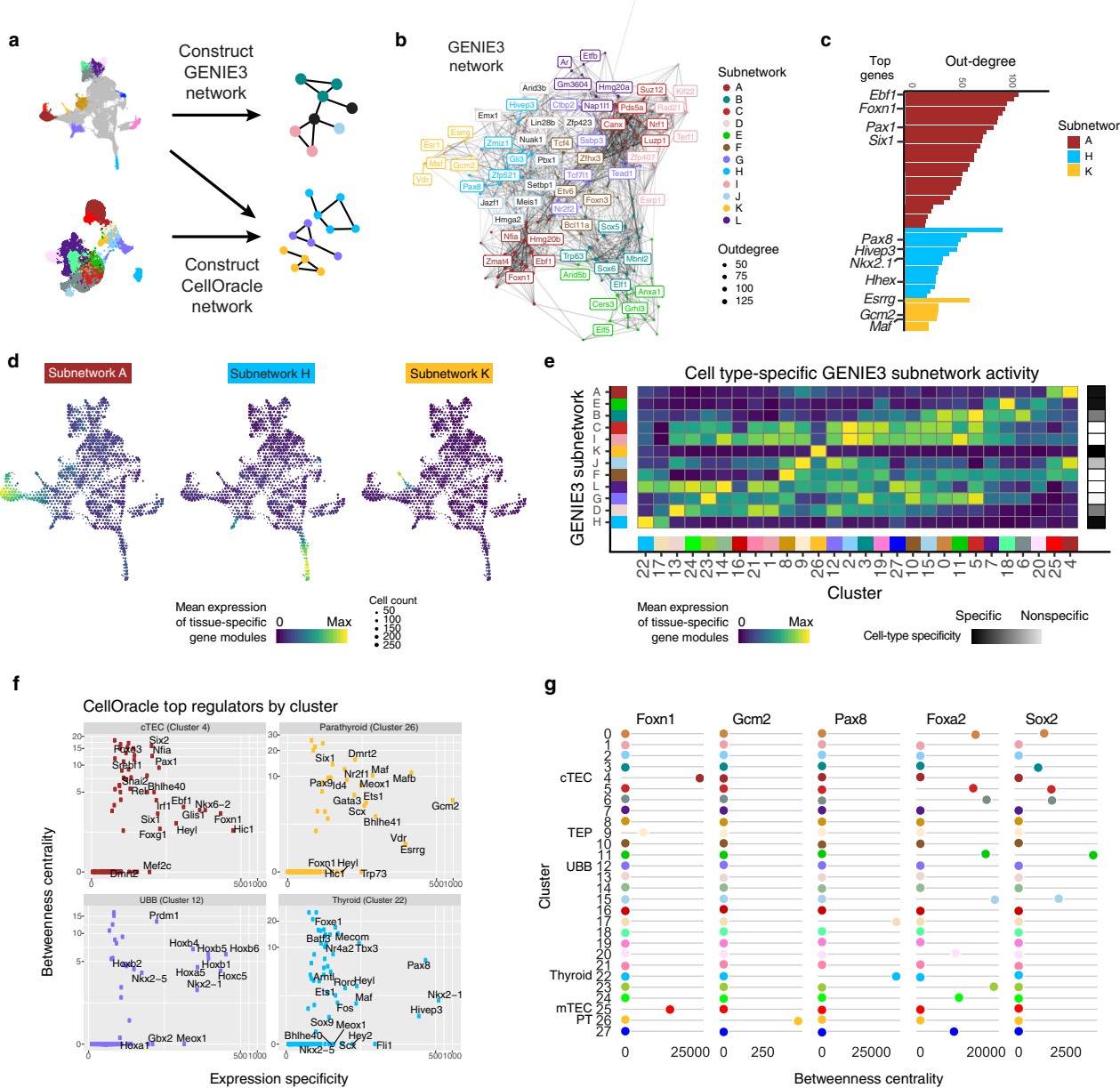

**Fig. 4 Gene regulatory networks reveal cell-type-specific regulatory interactions. a** Gene regulatory network (GRN) schematic for our two complementary approaches, GENIE3 and CellOracle. **b** GENIE3 GRN showing the top 200 regulators clustered into subnetworks. Regulators were filtered according to predicted regulation of the top differentially expressed genes across the entire dataset. The top 5 regulators of each subnetwork are labeled. **c** Regulators ranked by outdegree for three GENIE3 subnetworks: A, H, and K. **d** UMAP embeddings of the mean scRNA expression of subnetwork specific genes showing three highly cell-type-specific subnetworks. Subnetworks A, H, and K correspond to the thymus, thyroid, and parathyroid, respectively. Cells are binned by UMAP coordinate with size proportional to the number of cells and color signifying average log2-normalized expression. **e** Heatmap showing the cell-type specificity of GENIE3 subnetworks using the mean subnetwork activity on the scRNA atlas clusters. **f** Bivariate plot of average betweenness centrality from the cluster-specific CellOracle network and normalized expression specificity (average cluster-specific expression/median average cluster-specific expression) showing the top regulators per cluster. **g** Betweenness centrality network score dynamics as estimated by CellOracle across clusters for selected regulators of thymus (*Foxn1*), parathyroid (*Gcm2*), thyroid (*Pax8*), anterior foregut (*Foxa2* and *Sox2*) development. cTEC cortical thymic epithelial cell, TEP thymic epithelial progenitor, UBB ultimobranchial body, mTEC medullary thymic epithelial cell, PT parathyroid.

reduced cellularity in the *Foxn1* heterozygous thymus[60]. Closer evaluation of the cell-type abundance by cluster showed that the *Foxn1* KO is specifically depleted for clusters 4 and 25, which comprise the thymus lineage (Fig. 5d). Differential gene expression analysis between the *Foxn1* KO and control confirms all 5 differentially expressed transcripts previously identified by microarray profiling of E12.5 nude thymus: *Pdlim1*, *Mfsd12*, *Mreg*, *Fam57a*, and *Ppp1r16b*[61] (Supplementary Dataset 6).

By directly comparing the estimated transcriptomic state of the in silico *Foxn1* KO within the thymic clusters with our *Foxn1* KO experiment, we obtained a Spearman correlation of 0.32 (*p*-value < $2.2e^{-16}$) (Fig. 5e). Specific genes predicted in our network to be affected by a *Foxn1* KO and confirmed in our *Foxn1* KO experiment include those previously identified by Bleul et al. (*Ppp1r16b*, *Pdlim1*, *Mfsd12*, green labels)[61], additional genes implicated in thymus development, maturation, and function

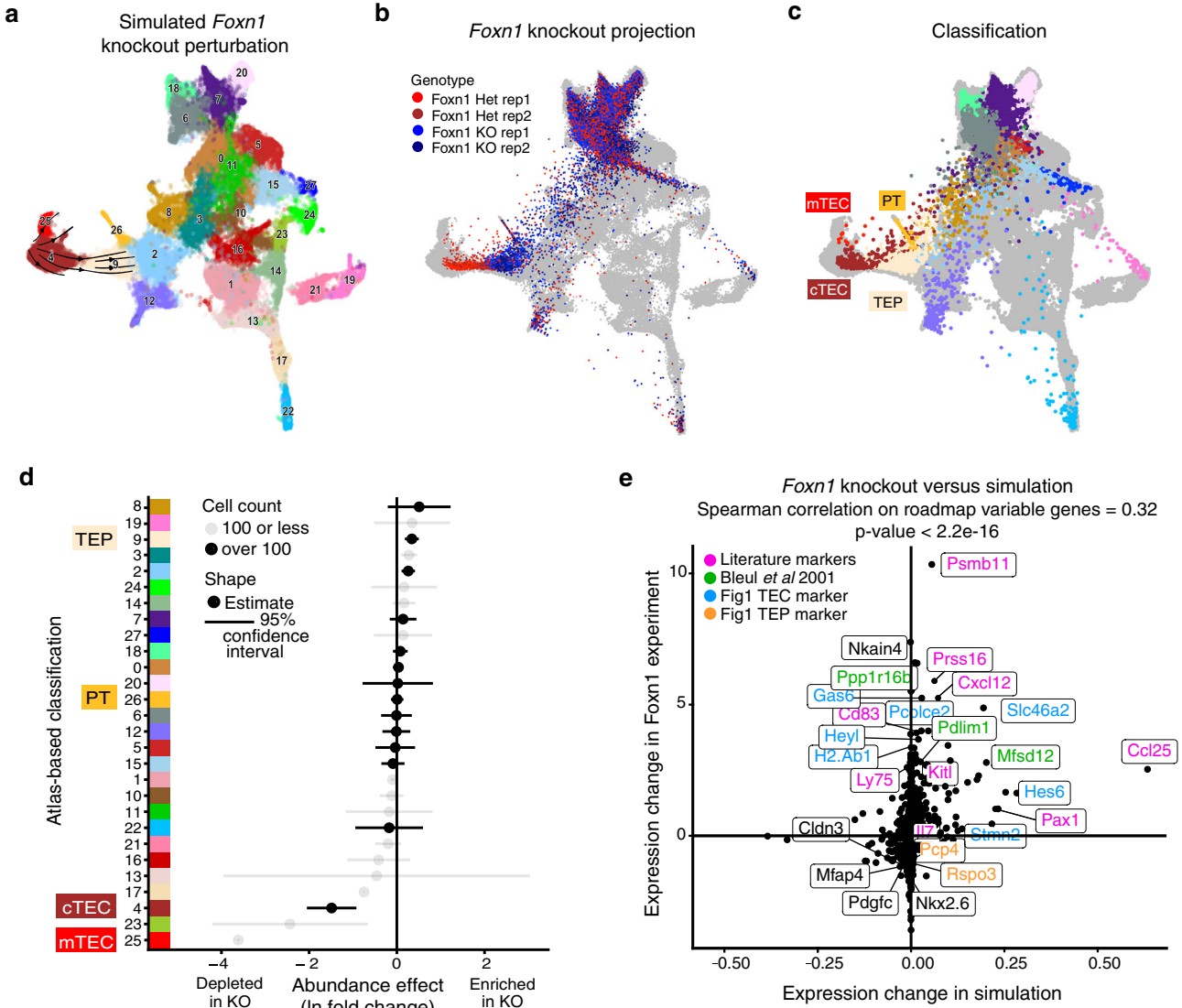

**Fig. 5 Foxn1 knockout validates GRN models of thymus development and shows transcriptomic effects highly similar to a developmental block. a** Visualization of an inducible *Foxn1* knockout as simulated by CellOracle. Changes in gene expression between the simulated knockout and atlas cells are displayed as velocity streams on the UMAP embedding using scVelo. **b**, **c** UMAP visualizations of *Foxn1* KO (n = 2) and heterozygous control (n = 2) data overlaid on the scRNA pharynx atlas. Each dot is one cell. Gray cells correspond to scRNA pharynx atlas. Colored cells are from the Foxn1 experiment, with color representing sequencing library (red, Foxn1 Het; blue, Foxn1 KO) (**b**) or classification (see Supplementary Fig. 3 for cluster color labels) (**c**). Classification and UMAP embedding of *Foxn1* KO cells was done via nearest neighbors. **d** Cell-type-specific abundance effects due to *Foxn1* KO. Clusters and colors are taken from the classifier results in (**c**). Dots show the log fold change in abundance, and lines show a 95% asymptotic confidence interval based on quasi-Poisson regression. Clusters with fewer than 100 total cells are shown in gray. **e** Gene-by-gene comparison of the estimated log2-fold change from the *Foxn1* KO experiment and the corresponding change predicted by the Foxn1 knockout simulation showing known and scRNA atlas identified thymus-specific genes.

(*Psmb11, Prss16, Cxcl12, Ccl25*, etc., pink labels)[62–65] as well as thymus markers from the TEC progenitors and cTEC clusters (9 and 4, respectively) in our transcriptomic atlas (*Rspo3, Gas6*, blue and orange labels; Fig. 5e, Supplementary Datasets 1 and 6).

To further assess the thymic-specific effects of *Foxn1* KO, we projected the *Foxn1* KO and control thymic cells onto a subset of the pharynx atlas only including third pouch progenitors and derivatives (Fig. 6a). The *Foxn1* experiment cells classified as E11.5 while the control cells classified as E12.5 (Fig. 6a, Supplementary Fig. 11). For a more refined measurement of these similarities, we determined the correlation between the *Foxn1* KO and third pouch TECs organized across pseudotime (Fig. 6b, Supplementary Fig. 11). This further demonstrated a

higher correlation between the *Foxn1* KO and E11.5 cells, whereas the control cells correlated more highly with E12.5 cells.

To determine the molecular signature of the observed developmental delay, we compared the gene expression differences between the *Foxn1* KO and control to the gene expression changes between E11.5 and E12.5 thymic cells (Fig. 6c, Supplementary Dataset 7). Overall, we detect 1218 differentially expressed genes between the *Foxn1* KO and control (q-value < 0.05) which have a 30% overlap with 47% of differentially expressed genes (absolute log2-fold change > 0.5, q-value < 0.05) between E11.5 and E12.5 (Supplementary Datasets 6 and 7). Many genes downregulated in the *Foxn1* KO that normally increase from E11.5 to E12.5 perform known roles in thymus

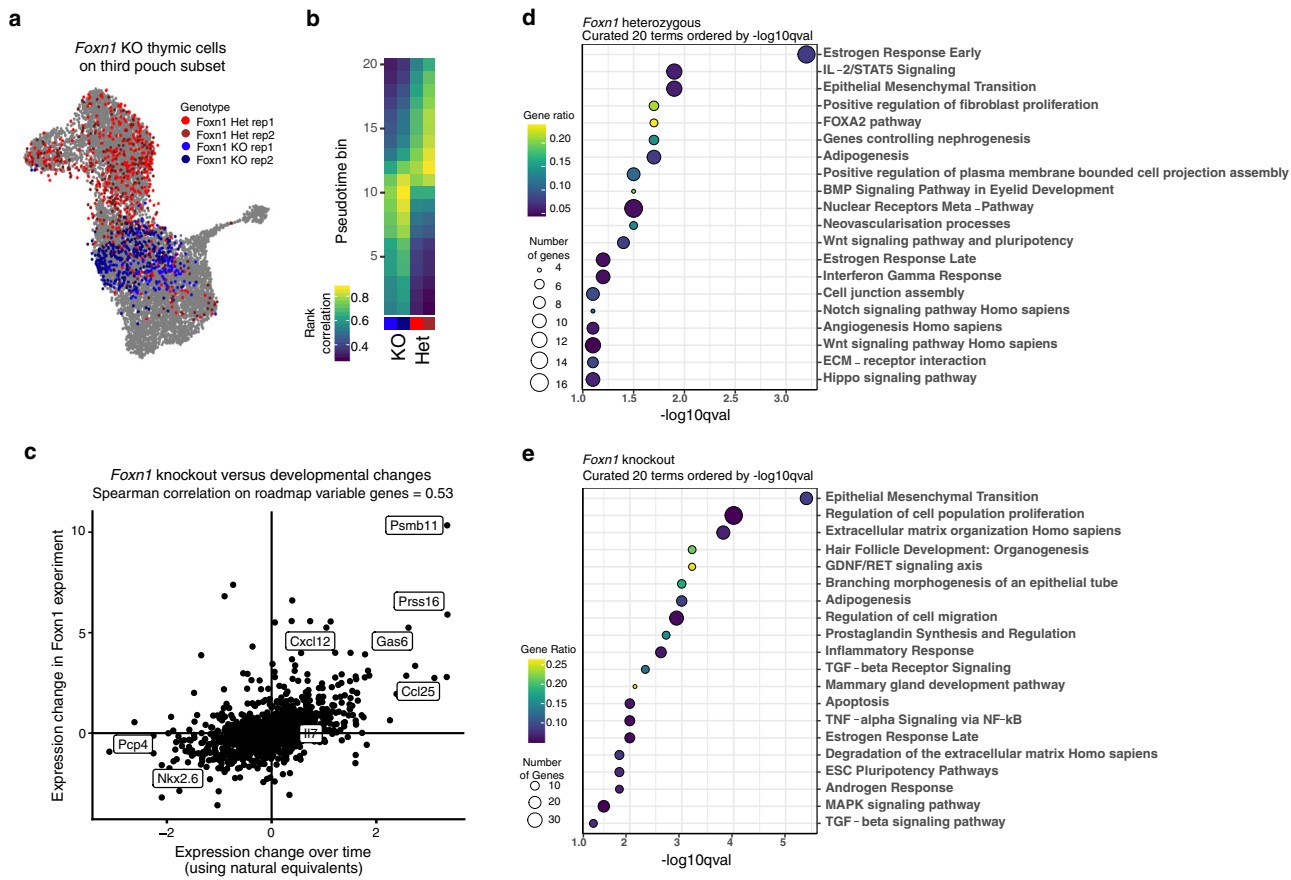

**Fig. 6 Characterization of Foxn1 deficiency reveals underlying molecular basis for developmental block. a** Projection of thymic cells from *Foxn1* KO (blue) and control (red) onto the third pouch subset analysis (gray), which is described in Supplementary Fig. 11. **b** Spearman correlation of Foxn1 experiment samples with pharynx atlas counterparts across pseudotime. Pseudotime is as shown in Supplementary Fig. 11 (DPT rank), and each bin contains an equal number of cells. Non-thymic cells from both datasets were omitted from this analysis. **c** Gene-by-gene comparison of *Foxn1* KO effects with "developmental block" model. Each dot represents one gene. Vertical axis shows log2-fold change between *Foxn1* KO and control. Horizontal axis shows log2-fold change in normalized expression between the scRNA atlas cells best correlated with the *Foxn1* KO (pseudotime bins 9–10) and with the control (pseudotime bins 12–13). **d, e** EnrichR analysis of genes differentially expressed between the control and *Foxn1* KO thymic cells. Dotplots contain curated lists of 20 terms enriched in the control (**d**) and *Foxn1* KO (**e**) thymus. Genes were selected based on a cutoff of log2-fold change > 0.5 and *q*-value < 0.01.

biology: *Dll4, Il7, Ccl25, Cxcl12*, and *Kitl* promote T cell proliferation and/or development[64,66,67], *Prss16* and *Ctsl* promote positive selection of CD4 single-positive T cells[65,68], and *Psmb11* promotes positive selection of CD8 single-positive T cells[63]. No significant effect on *Il7* was observed, consistent with existing models where *Il7* onset occurs independently of *Foxn1*[37].

To determine additional genetic programs that deviate from normal development in the *Foxn1* KO thymus, we analyzed differentially expressed genes (*q*-value < 0.01, log2-fold change > 0.5) for enriched gene ontologies. Genes with higher expression in the control thymic cells yielded terms related to various signaling pathways, including Wnt, BMP, and Notch, which have known roles in thymus development[49,69–72], as well as Hippo which has not yet been studied during early thymus development (Fig. 6d, Supplementary Dataset 8). We also find terms related to vascularization, a process known to be dependent on *Foxn1*[73] (Fig. 6d). Conversely, genes enriched in the *Foxn1* KO cells yielded terms including epithelial-to-mesenchymal transition (EMT), branching morphogenesis, and TGF-b signaling (Fig. 6e, Supplementary Dataset 8). Interestingly, *Foxn1* performs various roles in keratinocyte differentiation and wound healing via EMT[74], and a few EMT regulators are affected by the KO. *Snai2*, which turns on a keratinocyte re-epithelialization necessary for wound healing[75], increases from E11.5 to E12.5 and is downregulated in the *Foxn1* KO. Conversely, *Zeb2*, a

driver of EMT[76], is increased in the KO. Vimentin, a classic EMT marker[76], shows an increase in the knockout and is also downregulated from E11.5 to E12.5 (Supplementary Dataset 7). Several other genes normally upregulated during keratinocyte wound healing (*Tgfb2, Egfr, Pdgfc, Igfbp3*, and *Mapk11*) are also upregulated in the *Foxn1* KO[77]. These results support the hypothesis that *Foxn1* drives a re-epithelialization process in the thymus rudiment between E11.5 and E12.5, leading to increased *Snai2* and decreased *Zeb2* and vimentin expression.

Branching morphogenesis of an epithelial tube, a term enriched in the KO, stands out due to a recent study characterizing the morphological features of the nude thymus anlage[59]. Excitingly, our data provide possible genetic links between the loss of *Foxn1* and the branching morphogenesis phenotype reported by Muñoz et al. We detect increased expression of *Bmp2, Fgf10, Epha2, Sfrp2*, and *Slit2* among other branching morphogenesis-related genes, in the *Foxn1* KO (Supplementary Dataset 6). These genes normally function to promote branching morphogenesis in other epithelial organs including the mammary gland, kidney, and lungs[78–80]. While *Foxn1* deficiency during development leads to multifaceted and interrelated defects culminating in complete thymus dysfunction, our results begin to elucidate mechanisms behind morphogenic defects observed in the *Foxn1* KO thymus.

By examining the consequences of the *Foxn1* KO in the context of our transcriptomic time-course dataset, we further characterized the presumed developmental delay of the thymic primordium. In addition to confirming the downregulation of various genes related to thymus maturation in the *Foxn1* KO, we also described a potential EMT event occurring in the developing thymus, drawing similarities between the role of *Foxn1* in wound healing and thymus development, and providing putative effector genes associated with a branching morphogenesis phenotype.

## Discussion

Several human syndromes and disorders result from mal-development of the pharyngeal endoderm, which performs functions important for patterning and organization of the pharyngeal apparatus, and differentiation of organ-specific epithelial cells. Despite these important functions, the field lacks a comprehensive understanding of the molecular mechanisms underlying phar-yngeal endoderm development and, by extension, the causes of pharyngeal endoderm-associated syndromes and disorders. Here, we produced comprehensive single-cell datasets on the global transcriptomic and chromatin landscapes of developing phar-yngeal endoderm. We assessed heterogeneity both in terms of transcriptomic and chromatin accessibility signatures, thereby uncovering tissue-specific marker genes and putative CREs. For instance, we identified a putative early mTEC marker, *Grhl3*. While the field has relied on markers such as *Cldn3* and *Cldn4* to detect the earliest emergence of the mTEC lineage[39], our results demonstrate that such genes have far less specificity than *Grhl3*. Furthermore, we identified sets of cell-type-specific putative CREs, many of which are conserved across the Euarchontoglires clade of mammals. These results could identify non-coding elements contributing to the spectrum of human syndromes rooted in pharyngeal endoderm maldevelopment.

Using two complementary GRN approaches, we distill infor-mation on both global and cell-type-specific programs, leveraging both transcriptomic and chromatin accessibility information. These GRN results confirmed the central roles of various TFs known to regulate pharyngeal endoderm development and organ formation, and additionally identified other TFs with unknown roles in organ development. In the future, these GRNs can be refined with additional data types, such as ChIPseq to validate predicted TF targets. The GRNs defined here will also benefit from improved models for distinguishing between binding motifs within the same family, identifying de novo motifs, and studying combinatorial binding effects.

Based on a few predicted GRNs nodes, we simulated knockouts predicted to affect organ domains. We demonstrate the application of this resource for simulating and testing the consequences of loss of a key TF, *Foxn1*, during early thymus development. We inferred genes potentially responsible for a recently documented branching morphogenesis phenotype in the *Foxn1* KO thymus[59], which provides context on the molecular defects of a known primary immunodeficiency. Interestingly, Hippo signaling was active in the developing thymus and lacking in the *Foxn1* KO thymus, indicating that this pathway could be functioning downstream of *Foxn1* to regulate thymic proliferation and organ growth. This approach can now be extended to explore specific hypotheses about genes involved in development of the phar-yngeal endoderm, and consequently genes that could be con-tributing to the severe phenotypes of syndromes associated with defects in pharyngeal endoderm-derived organs. Although reg-ulatory redundancy or compensatory mechanisms could buffer penetrance of a factor knockout on GRN function and cellular state, such factors could be performing an important modifier function that would be apparent in the context of other genetic defects. This is especially relevant given that many syndromes associated with defects in pharyngeal endoderm development cannot fully be explained by known mutations and have variation in phenotypic penetrance.

Currently, the GRN models can only model defects in phar-yngeal endoderm development and, integration with datasets including additional pharyngeal cells could reveal how defects in one cell type can propagate in the surrounding tissue. Finally, given the importance of pharyngeal endoderm epithelia in pharyngeal organ function, this resource can inform and benchmark directed differentiation of pluripotent stem cells to pharyngeal endoderm derivatives for the purposes of disease modeling and cell replacement therapies[81]. Overall, this resource has diverse applications across several fields and provides a greater understanding of the molecular processes underlying pharyngeal endoderm development between pouch formation and early organogenesis.

## Methods

**Mouse husbandry and embryo collection.** All mouse experiments were per-formed in accordance with the regulatory standards defined by the National Institutes of Health and the University of Massachusetts Chan Medical School Institutional Animal Care and Use Committee (IACUC), protocol #2384. C57BL/6J mice (000664), *Foxn1*[nu] (B6.Cg-*Foxn1*[nu]/J; 000819) were obtained from The Jackson Laboratory. C57BL/6J females between 6 and 8 weeks of age were crossed with *Pax9*[VENUS] genetically targeted reporter mice[19]. Heterozygous *Pax9*[VENUS] embryos (identified by visual inspection for expression VENUS protein) were dissected at timepoints embryonic day (E)9.5, E10.5, E11.5, and E12.5, where the morning of plug detection was considered E0.5. Dissections were performed in cold RF10-H medium (RPMI 1640 (Gibco; 22400089) supplemented with 2 mM L-glutamine (Gibco; 25030081), 100 IU/mL penicillin and streptomycin (Corning; 352063), and 10% v/v Gibco fetal bovine serum (Gibco; 10437028)). For single-cell RNA-seq and single-cell ATAC-seq assays, pharyngeal endoderm was isolated from heterozygous *Pax9*[VENUS] embryos as described below. To isolate *Foxn1*[nu/nu] pharyngeal endoderm, *Foxn1*[nu/wt]*Pax9*[VENUS/wt] mice were crossed with *Foxn1*[nu/wt] female mice between 6 and 8 weeks of age. All embryos resulting from crosses with *Foxn1*[nu/wt] mice were genotyped as described below prior to cell isolation and FACS.

**Thymocyte isolation and flow cytometry.** Thymocytes from 5-week-old litter-mates (wild type, *Foxn1*[nu/wt], and *Foxn1*[nu/wt]*Pax9*[VENUS/wt]) were obtained by mechanical agitation of the thymus. Cells were incubated with anti-mouse CD16/ 32 (Biolegend; 101302) for 10 min prior to staining with APC anti-mouse CD4 (BD Pharmingen; 553051) and PE anti-mouse CD8a (BD Pharmingen; 553033). Dead cells were excluded with 7-AAD (BD Biosciences; 51-68981E). Data were acquired using a BD Accuri C6 and processed using FlowJo [v10.8.0].

**Embryo genotyping for the presence of *Foxn1*[nu] allele.** DNA was extracted from embryo limbs using the EZ Fast Tissue/Tail PCR Genotyping Kit (EZ Bio Research; G1001-300). DNA was extracted in 50 μL of DNA extraction solution and diluted 1:10 prior to downstream genotyping. Presence of the *Foxn1*[nu] allele was assayed via restriction digest as specified in The Jackson Laboratory Foxn1[nu] protocol (000819). Briefly, genomic DNA was amplified using oIMR1292 (5′ GGC CCA GCA GGC AGC CCA AG) and oIMR1293 (5′ AGG GAT CTC CTC AAA GGC TTC) primers. Following PCR purification using the MinElute PCR Purification Kit (Qiagen; 28004), PCR products were digested with FastDigest BseDI (Thermo Scientific; FD1084) at 37 °C for 10 min. Restriction digest products were resolved on a 5% agarose gel.

**Cell isolation and FACS.** The pharynx region was mechanically dissected from heterozygous *Pax9*[VENUS] embryos through removal of the head and lower torso from above the heart. Tissue was washed in cold 1X PBS, then dissociated in 0.05% trypsin-EDTA (GIBCO; 25300-120) and DNase (Sigma; DN25-1g) at 37 °C for 15–20 min with P1000 pipette trituration every 5 min. Trypsin was deactivated with RF10-H medium. Single-cell suspensions were filtered through 40 μm cell strainers (Fisher Scientific; 22363547) and centrifuged at 4 °C, 300 × *g* for 5 min. Cells were treated with 1X RBC lysis buffer (eBioscience; 00-4333-57) at 4 °C for 3 min, then washed with FACS buffer (1X PBX + 2% v/v FBS). Cells were stained with a PE/Cy7 anti-mouse CD326 EpCAM antibody (Biolegend; 118216; 1:1000) in FACS buffer at 4 °C for 20 min. Dead cells were excluded with 7-AAD (BD Biosciences; 51-68981E). Cell sorting was performed byf the University of Massachusetts Chan Medical School Flow Cytometry Core using a BD FACS S-Aria IIu Cell Sorter 5B 2V 2R or a BD FACS 2-Aria IIu Cell Sorter 2B 6V 3R 5YG, and approximately 20,000 *Pax9*[VENUS] and PE/Cy7 double-positive cells were collected per replicate. Cells were sorted into 1.5 mL Eppendorf tubes pre-coated with FACS buffer.

**Single-cell RNA-seq library preparation.** Following FACS, all ~20,000 cells were pelleted at $300 \times g$ for 5 min and resuspended in 33.8 µL or 46.6 µL of FACS buffer according to the Chromium Single Cell 3′ v2 and v3 chemistry kits, respectively (10X Genomics; CG00052 Rev D; CG000183 Rev A). Total cell suspensions were combined with Master Mix (10X Genomics; 120237; 1000092) and loaded on a Chromium Controller. Libraries were generated using Chromium Single Cells 3′ Reagent Kits according to manufacturer's instructions (10X Genomics; 120237; 1000092).

**Nuclei isolation for single-cell ATAC-seq and library preparation.** Between 30,000 and 40,000 cells were sorted, then centrifuged at 4 °C, $300 \times g$ for 5 min. To isolate nuclei, 100 µL of chilled lysis buffer (10 mM Tris-HCl, 10 mM NaCl, 3 mM MgCl$_2$, 0.1% v/v Nonidet P40) was added to the cell pellet, which was then transferred into a 0.2 mL PCR tube. Nuclei were centrifuged at 4 °C, 850 rcf for 5 min, then washed once in 1X PBS with 0.04% BSA. Based on the FACS cell count, nuclei were resuspended in an appropriate volume of Diluted Nuclei Buffer (10X Genomics; CG000168 Rev B, 2000153) to achieve a targeted nuclei recovery of ~6000. Libraries were generated according to manufacturer's instructions (10X Genomics Chromium Single Cell ATAC Reagents Kits; 1000111, 10000086, 1000084).

**Next-generation sequencing.** The average base-pair size for each library was determined using an Agilent High Sensitivity DNA Bioanalyzer or an Advanced Analytical Fragment Analyzer, and final library concentration was determined using a Qubit High Sensitivity DNA assay kit (Invitrogen; Q32854). All libraries were sequenced on a NextSeq 500. Pharynx atlas single-cell RNA-seq libraries were sequenced using a 75 cycle NextSeq 500/550 High Output Kit v2 (Illumina; FC-404-2005) at 26(Read 1)/50(Read 2). All single-cell RNA-seq libraries pertaining to the *Foxn1*$^{nu}$ mice were sequenced using a 75 cycle NextSeq 500/550 High Output Kit v2.5 (Illumina; 20024906) at 26(Read 1)/50(Read 2). Single-cell ATAC-seq libraries were sequenced using a 150 cycle NextSeq 500/550 High Output Kit v2.5 (Illumina; 20024907) at 65(Read 1)/65(Read 2). In addition, libraries were indexed for multiplexed sequencing.

**Embryo fixation, sectioning, and RNA in situ hybridizations and immunofluorescence.** Embryos were fixed in 10% Neutral Buffered Formalin (Sigma-Aldrich; HT5012) for 16 h/overnight. Embryos were ethanol dehydrated and paraffin-embedded, then sectioned at the University of Massachusetts Chan Medical School DERC Morphology Core. All paraffin blocks were cut at 5 µm per section. Fluorescence in situ hybridization was performed using the RNAscope Multiplex Fluorescent Reagent Kit v2 Assay according to manufacturer's instructions (ACD Bio; 323100). The following RNAscope probes were used (all purchased from ACD Bio): Foxn1 (482021), Gas6 (450941), Grhl3 (540461), Gcm2 (530481), Sparcl1 (424641), Ibsp (415501), Ngfr (494261), Flrt2 (490291), Calca (417961), Ascl1 (313291), Meox1 (530641). Probes were fluorescently tagged using Opal dyes (Perkin Elmer; FP1488001KT, FP1497001KT). Sections were mounted with Prolong Gold Antifade Mountant (Thermo Fisher Scientific; P36930).

For immunofluorescence stains, paraffin-embedded tissue was deparaffinized and rehydrated, then subjected to antigen retrieval in Sodium citrate buffer (10 mM Sodium citrate, 0.05% Tween 20, pH 6.0) for 30 min at 95 Celsius. Tissue was blocked with 0.5% Donkey Serum in PBS-T (1X PBS with 0.2% Tween 20), then incubated overnight at 4 Celsius with primary antibodies anti-Epcam (Biolegend; 118202; 1:1000) and anti-Pax9 (Santa Cruz Biotechnology; sc-7746; 1:100) diluted in blocking buffer. After 3 washes for 5 min in PBS-T, sections were incubated with secondary antibodies Alexa Fluor 594 donkey anti-rat (Invitrogen; A21209; 1:300) and Alexa Fluor 488 donkey anti-goat (Invitrogen; A11055; 1:300) for 1 h at room temperature. After 3 washes for 5 min in PBS-T, sections were incubated with Hoechst 33342 (Invitrogen; H3570) for 15 min at room temperature. Slides were washed 3 times for 5 min with PBS, then mounted using Fluoromount-G mounting medium (Thermo Fisher Scientific; 00-4958-02). All in situ and immunofluorescence samples were imaged on a Zeiss LSM 900+Airyscan Microscope using Zeiss ZEN [v3.1 (blue edition)].

**Alignment, quantification, and statistical analysis of single-cell RNA data.** FASTQ creation, alignment, and quantification were performed using 10X Cell-ranger (version 2.1.0) with an mm10-based default reference (version 1.2.0). Downstream work predominantly used the R package Seurat[82] (version 2.3.3), with additional utilities from the R package thymusatlastools2 (https://github.com/maehrlab/thymusatlastools2).

**Single-cell RNA atlas quality control and filtering.** Two samples at E10.5 were excluded due to low depth, leaving the atlas with a total of 57850 cells and ten samples (E9.5: 2; E10.5: 3; E11.5: 3; E12.5: 2). Predicted doublets were marked (but not removed) using the simulation-based method doubletFinder[83]. Numbers of expected doublets were calculated by linearly interpolating 10X's published table of doublet percentages versus final cell count. In total, 2789 predicted doublets were marked.

Counts were rescaled to 10,000 per cell and log2-transformed with a pseudo-count of 1. A typical Seurat workflow was followed, using gene selection (with

FindVariableGenes), scaling (with ScaleData), and principal components analysis (PCA[84] with RunPCA), graph-based clustering with the Louvain algorithm[85] with FindClusters, and UMAP visualization[86] with RunUMAP. For this initial analysis, 967 genes were selected (expression cutoff: 0.1; dispersion cutoff: 0.8). 100 principal components (PCs) were retained for downstream steps, and the Louvain graph clustering (via FindClusters) used a resolution of 2. Out of the resulting 42 clusters, we removed:

- Cluster 33 (459 cells) due to high Neurod1 and low Pax9 (associated with neuronal signature[87])
- Cluster 41 (79 cells) due to high Pax3, and low Pax9 and Epcam (associated with neural crest signature[88])
- Cluster 36 (381 cells) due to high Esam and Pdgfra, moderate Dlx5, and low Pax9 and Epcam (associated with endothelial signature[89])
- Cluster 38 (247 cells) due to high Pax3 and low Pax9 and Epcam (associated with neural crest signature[88])
- Cluster 39 (231 cells) due to high Rxrg, Dlx5, and Sox10, and low Pax9 (associated with otic vesicle signature[90–92])
- Any cell with over 7% mitochondrial UMIs (2122 cells)
- Any cell with under 1% mitochondrial UMIs (270 cells)
- Any cell with log normalized Hbb-bt expression over 1 (17 cells) (associated with red blood cell signature)

This brought the total cell count down to 54,044 (5734 and 7611 cells from E9.5 replicate 1 and 2, respectively; 4743, 4129, and 4248 cells from E10.5 replicate 1, 2, and 3, respectively; 5385, 3141, and 7967 cells from E11.5 replicate 1, 2, and 3, respectively; 4885 and 6201 cells from E12.5 replicate 1 and 2, respectively).

**Single-cell RNA atlas overview analysis.** The 54,044 cells passing quality control were analyzed using Seurat to produce overview figures, 1119 genes were selected (expression cutoff = 0.1; dispersion cutoff = 0.8). Following the tactic of Farrell et al.[93], 64 PCs were retained based on the Marchenko-Pastur upper bound[94] for UMAP and clustering. The Louvain graph clustering used a resolution parameter of 1.

The gene selection and centering/scaling were both modified to incorporate a regression-based batch correction procedure to suppress variation due to sequencing depth, cell cycle, and batch effects. This begins with a matrix of size $n$ by $d$, where $n$ is the number of cells and $d = 10$ is the number of variables involved in the batch correction. The variables are UMI counts (1 column), cell cycle phase indicators (3 columns), and contrasts between replicates within each timepoint (6 columns; for example, a vector containing 1 for cells from E12.5 rep1, −1 for E12.5 rep2, and 0 for the rest, with certain redundant columns omitted because they lie in the linear space spanned by the included columns). This was implemented using the function get_batch from thymusatlastools2, a R package of utility functions whose latest release accompanies this project. For the gene selection, Seurat typically uses normalized counts (but without pseudocounts or log transformation) to calculate mean and variance, and it selects genes with variance higher than expected given the mean. Instead of the variance of the normalized counts, the variance of their residuals after a least-squares multiple regression on the aforementioned factors. This was implemented using the function GetDispersionCalculator from thymusatlastools2. Uncorrected normalized counts are passed on to the next step, centering and scaling. For this step, the Seurat workflow typically starts with log2-transformed normalized counts, from each gene subtracting the mean and dividing by the standard deviation. Instead of subtracting the mean, we subtracted the fitted values from least-squares regression. The resulting values are used in PCA. This was implemented via Seurat's ScaleData function via the argument vars.to.regress.

The top differentially expressed genes for Supplementary Fig. 3 (top 10) and Supplementary Dataset 1 (top 100) were obtained using the Seurat function FindMarkers (test.use = 'mast') for each cluster over a background comprising the remaining cells in the dataset. FDR corrected $p$-values were obtained using the R function p.adjust. The resulting genes were ranked by the ratio of the percentage of cells in which the gene was detected in that cluster to the detection percentage in all other cells. The top differentially expressed genes for each cluster in Fig. 1e were obtained using Seurat's FindAllMarkers (test.use = "bimod") with a $p$-value cutoff of 0.01 and log2-fold-change cutoff of 1.5.

**Single-cell ATAC atlas data processing and quality control.** De-multiplexing of scATAC data was performed with Cellranger-ATAC version 1.0.0 (atlas E12.5 rep1,2 and E11.5 rep1) and 1.1.0 (atlas E11.5 rep2) with an mm10-based default reference (refdata-cellranger-atac-mm10-1.0.0 from 10x Genomics). Downstream work predominantly used the ArchR package[95] (ArchR_1.0.1 in R 3.6.3 (2020-02-29)) with additional utilities as described.

To remove cells with low sequencing depth and low signal-to-noise ratio, cells with <3000 unique fragments or TSS enrichment ratio <4 were filtered out leaving 4320, 1273, 3576, and 4784 cells from E11.5_rep1, E11.5_rep2, E12.5_rep1, and E12.5 rep2 respectively. Gene expression scores were computed using ArchR default method ("Model 42") which uses accessibility in 500 base-pair windows within the gene body (upto 5 kb upstream of the TSS) and the weighted accessibility in distal regions (100 kb on either side of the gene body, excluding the gene body region) not overlapping other gene regions to infer gene expression. Per sample doublet enrichment was computed on the remaining 13,953 cells using ArchR addDoubletScores with default parameters. ArchR filterDoublets was run

per sample to remove the $\frac{\text{number of cells}^2}{10000}$ cells with the highest doublet enrichment removing 186, 16, 127, and 228 cells from E11.5_rep1, E11.5_rep2, E12.5_rep1, and E12.5 rep2, respectively, leaving 13,396 cells. Initial unsupervised analysis included computing fragments counts in 500-base-pair non-overlapping tiles, feature selection and iterative LSI (2 iterations, no downsampling, 12 clusters for pseudobulk identification of 25,000 variable features, dimensions = 35), UMAP embedding (30 nearest neighbors, minDist = 0.2) and Louvain clustering (resolution = 4, via the interface to Seurat) to identify 42 clusters (Supplementary Fig. 4). Cells from the pre-cleaned scRNA atlas (Supplementary Fig. 1) were assigned cluster labels 0 to 27 based on their cluster in the scRNA atlas (Supplementary Fig. 3) and the label "removed" if they were identified as a contaminant population. In order to identify non-endodermal scATAC populations, this scRNA dataset was integrated with the scATAC data using ArchR's addGeneIntegrationMatrix (no downsampling, 30 LSI dimensions, top 2000 variable genes) which performs canonical correlation analysis using Seurat to learn a joint embedding on the imputed scATAC gene scores and scRNA gene expression and assigns each scATAC cell an scRNA cluster label and normalized gene expression signature to each scATAC cell based on the nearest scRNA cell in the integrated space. scATAC clusters C40, C42, C1, C37, C41, C28, and C36 were thus identified as non-endodermal. scATAC clusters C30, C11, C31, C29, and C27 contained cells mapping to multiple scRNA clusters and were identified as putative doublets. The median doublet enrichment of these clusters was >1 standard deviation away from the average of the median doublet enrichment across all the scATAC clusters confirming their doublet status. In total, 12 clusters (7 non-endodermal, 5 doublet) were removed leaving 10,890 scATAC cells across 4 samples as follows: E11.5 replicate 1 = 3248, E11.5 replicate 2 = 1075, E12.5 replicate 1 = 2813, and E12.5 replicate 2 = 3754.

**Single-cell ATAC atlas analysis.** Feature selection and dimensionality reduction using iterative LSI (2 iterations, no downsampling, 12 clusters for pseudobulk identification of 30,000 variable features, dimensions = 30) were performed on the scRNA atlas cells. Two dimensions correlated (correlation > 0.4) with the sequencing depth were removed from all downstream analysis except for integration with the scRNA data. The UMAP embedding (30 nearest neighbors, minDist = 0.5), Louvain clustering (resolution = 1.4) and integration with the scRNA atlas cells from day E11.5 and E12.5 (same parameters as before) was performed to generate the overview figures.

Pseudobulk coverage was generated from the 22 clusters using ArchR addGroupCoverages (minimum cells in a replicate = 40, maximum cells in a replicate = 500, minimum replicates per cluster = 2, sampling ratio = 0.8, kmerLength for Tn5 bias correction = 6). Clusters C1–9, C11-13, C15, C19-22 gave 2 replicates each and the remaining cluster gave 3 replicates. 501-base-pair fixed width peaks (250 bp on either side of the summit) were called separately on each pseudobulk replicate using the ArchR interface to MACS2 (maximum peaks per pseudobulk replicate = 500 times the number of cells or 150,000, whichever is smaller, excluding chromosomes M and Y, MACS2 significance $q$-value cutoff = 0.1) and merged using ArchR iterative overlap method. In brief, overlapping peaks from different pseudobulk replicate in the same cluster were merged by retaining the peak with the highest normalized significance score and peaks occurring in at least 2 pseudobulk replicates were retained. Finally, overlapping peaks were merged across clusters yielding 270,210 peaks. Peaks were annotated as promoter (any overlap with regions upto 2000 bp upstream and 100 bp downstream of a TSS) exonic (any overlap with exonic genic region), intronic (any overlap with genic regions that are not exonic), and distal (all remaining peaks). A "number of cells by number of peaks" PeakMatrix was generated by counting the number of insertions per peak per cell upto a maximum of 4 (counts > 4 are set to 4) to prevent bias. Differentially accessible cluster-specific peaks were found by a Wilcoxon test against a background set (using TSSEnrichment and log10-nFrags as bias variables, $k = 100$ neighboring cells, minimum fraction of cells in background set = 0.8) using ArchR getMarkerFeatures (maxCells per cluster = 500) on the PeakMatrix (scaled to 10,000). Differentially accessible peaks (FDR ≤ 0.01, log2-fold change ≥ 0.5) which were annotated as distal or promoter, respectively.

Browser tracks were generated using ArchR plotBrowserTrack with bulk ATAC signal per cluster (insertions summed in bins of 250 base pairs, normalized by reads in the TSS), peak regions and line diagrams of genic regions (blue indicating genes on the minus strand and red indicating the plus strand) centered at the TSS of the respective genes and extending 50 kb up and downstream.

The UMAP embeddings in Fig. 2 were generated by imputing the respective signal using the ArchR interface to MAGIC[96] (with scaling of the Iterative LSI dimensions excluding 2 dimensions correlated at >0.4 with sequencing depth, diffusion time parameter td = 3, k-nearest neighbors for smoothing = 5, sampleCells = 5000, kNN autotune parameter ka = 4, number of imputation replicates = 2, standard deviation for kernel = 1).

ArchR's getMarkerFeatures with the Wilcoxon test corrected for TSSEnrichment and log10(nFrags) was used to identify differential (FDR ≤ 0.1, log2-fold change ≥ 0.5) GeneScores per cluster over the remaining cells for Supplementary Dataset 2 and the top 50 were visualized in Fig. 2f using ArchR's plotMarkerHeatmap function. The top 2 genes per cluster plus an additional curated set of markers were labeled on this heatmap. The differentially accessible

peaks were obtained in a similar manner for Fig. 3 (mean signal per cluster, log2-transformed, scaled to 10,000, row z-scored, upper limit = 2, lower limit = −2, binary clustering on peaks, columns grouped by dendrogram). Promoter and distal differential peaks with FDR ≤ 0.1, log2-fold change ≥ 0.5 were presented in the Supplementary Dataset 3. The peaks with FDR ≤ 0.01, log2-fold change ≥ 0.5 were visualized in the heatmap with a customized version of plotMarkerHeatmap for differentially accessible promoter peaks to enable explicit ordering of the clusters.

Cell-type specificity for the 27,295 distal and 2436 promoter peaks was computed using the scaled log2-transformed heatmap matrices (without z-scoring and with no upper or lower limits). For each peak, the cluster specificity index $\tau = \frac{\sum_{i=1}^{N}(1-x_i)}{N-1}$ was determined as described in Yanai et al.[97], where $N$ is the number of clusters ($N = 22$) and $x_i$ is the scaled log2-transformed mean signal for a given peak in cluster $i$. The cluster-specificity index distribution from promoter and distal peaks was visualized using ggplot and a two-sided Wilcoxon test was performed using the R function wilcox.test.

Cis-BP version 2.00[98] database for Mus musculus was downloaded from http://cisbp.ccbr.utoronto.ca/bulk.php. All motifs were loaded into R using the universalmotif (https://bioconductor.org/packages/release/bioc/html/universalmotif.html) and TFBSTools[99]. The resulting motif PWMList was added to the ArchR project using addMotifAnnotations (cutoff = 5e−05, width = 7, version = 2) to obtain a binary peak-motif matrix. For each peak, a background set of 50 peaks controlled for accessibility and GC-content was computed using addbgdPeaks ($w = 0.1$, binSize = 50). Bias-corrected deviation scores (z-scores of deviation across all cells) in single-cell motif accessibility from the expected accessibility across cells were computed with ArchR addDeviationsMatrix which uses a scalable implementation of ChromVar[100] and visualized on UMAP embeddings in Fig. 2.

An ArchR project with the set of 55,840 differentially accessible distal peaks (FDR ≤ 0.1, log2-fold change ≥ 0.5) was created and peaks and motifs were added to it in the same manner as before. Correlation between the gene integration and motif deviation score matrix was computed for every motif-gene pair in the same family using corbetw2mat from the lineup package[101]. The per cluster motif score for a motif was defined as difference between the average motif deviation score of that cluster and the minimum average deviation score across all clusters. Genes-motif pairs were obtained by selecting the top motif score per gene and the top correlated gene per motif. Finally, pairs that had correlation >0.35 and motif scores in the top-10 percentile for that cluster were retained and plotted using scanpy's matrixplot function for the motif score and seaborn's heatmap function for the correlation color bar in Fig. 3c. Motif logos were visualized by converting the PWMatrix to probabilities with TFBSTools[99] and using a modified version of the seqLogo function from https://bioconductor.org/packages/release/bioc/html/seqLogo.html to add sequence logos[102].

Co-accessibility between peaks upto 250 kb apart was obtained using ArchR's addCoAccessibility on the 28 scaled iterative LSI dimensions (2 dimensions were excluded due to >0.4 correlation with sequencing depth). Briefly, 491 low-overlapping aggregates of cells ($k = 100$, knnIteration = 500, overlapCutoff = 0.8) were obtained using an optimized version of the Cicero approach for sampling and aggregation of cells[103] and the correlation between their log2-normalized accessibility was computed. Correlated peak pairs were removed if they had high FDR (≥10^−10), if either of the peaks had low variability (≤0.35) across the dataset or if the correlation was low (<0.5) leaving 164,417 peak pairs which were used in the gene regulatory network.

**Identification of conservation scores.** PhastCon scores[104] for the Euarchontoglires clade were downloaded from http://hgdownload.cse.ucsc.edu/goldenpath/mm10/phastCons60way/mm10.60way.phastCons60wayEuarchontoGlire.bw. Distal peaks from Fig. 3a were filtered into those with an average phastCon score of >0.5. Differentially accessible and conserved peaks for each cluster were submitted to GREAT[105] using the rGREAT R package (http://bioconductor.org/packages/devel/bioc/html/rGREAT.html). For each curated term, the binomial FDR value was selected. Heatmaps were generated using scanpy[106].

**Gene regulatory network inference.** Prior to GRN inference, the 54,044 cells were aggregated into 555 metacells via the metacell R package[107]. Gene selection via mcell_gset_filter_varmean and mcell_gset_filter_cov used parameters T_vm = 0.08, T_tot = 100, and T_top3 = 2. For KNN construction, mcell_add_cgraph_from_mat_bknn used K = 100. For the resampling step, mcell_coclust_from_graph_resamp used parameters min_mc_size = 20, p_resamp = 0.75, and n_resamp = 500, and for the final step, mcell_mc_from_coclust_balanced used parameters mc_id = "test_mc", K = 30, min_mc_size = 30, and alpha = 2.

Metacells were then passed to SCENIC[108]. Genes were filtered, requiring a minimum total count of 10, a minimum of 2 metacells in which each gene is detected, and requiring presence in the RcisTarget gene databases mm10__refseq-r80__500 bp_up_and_100 bp_down_tss.mc9nr (mm10, 500 bp upstream of TSS, and 100 bp downstream) and mm10__refseq-r80__10 kb_up_and_down_tss.mc9nr (mm10,10kb up and down of TSS) leaving 15,329 genes. GENIE3 from the SCENIC package was run and edges with weight ≥0.01 were retained. To identify key regulators, we began with a set of 1240 genes whose variance was best explained by the pharynx scRNA atlas principal components. For each regulator predicted by

GENIE3, targets were intersected with this list, and regulators were ranked by the size of the intersection. The subgraph containing the top 200 regulators converted to an adjacency matrix, clustered using the leiden algorithm (resolution = 2) and displayed in 2D using the igraph R package[109]; functions make_undirected_graph, as_adjacency_matrix, layout_with_fr) and the leiden package[110] (https://github.com/TomKellyGenetics/leiden; function leiden). For each cluster, all regulators were averaged and shown on featureplots using save_feature_plots (mode = overplot_adjust) from thymusatlastools2.

**Network inference with CellOracle**. GRN inference and Foxn1 KO simulation was carried out using a combination of motif analysis in open chromatin and correlative analysis of scRNA data. Atlas scRNA and scATAC datasets were used; no Foxn1 KO data was used.

*Base GRN construction*. Peaks and peak-to-peak co-accessibility were obtained in ArchR. A peak was associated with a target gene if it overlapped with the TSS (TSS peak, co-accessibility = 1) or if it had a co-accessibility ≥0.5 with a TSS peak. Peaks were scanned for motifs using CellOracle's scan function which uses the gimmemotifs[111] motif scanner (background_length = 200, fpr = 0.02, default motif database = gimme.vertebrate.v5.0 using binding and inferred motifs and cumulative binding score cutoff = 10) to generate an annotated peak-motif binary matrix which is the base GRN in CellOracle.

*Cell-type-specific GRN refinement and simulation*. Using CellOracle, the base GRN was refined using the atlas scRNA data to form cell-type-specific GRN's and simulate the Foxn1 KO. 10,000 genes were selected by requiring at least 3 counts and by using scanpy's preprocessing utility 'sc.pp.filter_genes_dispersion' with 'flavor = 'cell_ranger' and 'n_top_genes = 10,000'. The selected genes included all 1119 variable genes used in the atlas scRNA analysis. Normalized (using sc.pp.normalize_per_cell) count data was imputed in a 50 principal components subspace using CellOracle's balancedKNN implementation ($k = 54$ nearest neighbors, b_sight = 54*8, b_maxl = 54*4). Cell-type-specific GRN's were trained using CellOracle's default procedure: for each target gene and each cell type, bagging ridge regression was run (bagging_number = 20, alpha = 10) using connections determined by the base GRN. Edges were preserved with edge $p$-value ≤ 0.001 where the source node was in the top differentially expressed genes of that cluster as determined by Seurat's FindAllMarkers (log2-fold change threshold = 0.25, return.thresh = 0.01) on the scRNA atlas and the edge weight was >0.005 leaving us with a trained GRN. Network statistics including betweenness centrality were computed for each GRN using CellOracle's getscore function. GRNs were re-trained on the preserved edges using the same parameters as before. For the knockout simulation, for each network containing Foxn1, the Foxn1 expression was set to 0 and propagated through the network upto a depth of 5.

**Analysis of the simulated Foxn1 knockout**. To investigate the shift in gene expression in the presence of the Foxn1 knockout simulation, we visualized the direction and magnitude of the simulated knockout using stream plots in scVelo[112] (scvelo 0.2.2.dev51 + ga7de78a (python 3.6.10)) since transition probabilities could not be computed in CellOracle on the atlas scRNA data (54,044 cells) due to high memory requirements. The k-nearest-neighbor graph was obtained on the scRNA atlas principal component space using scvelo's neighbors (n_neighbors = 30, metric = 'euclidean') function. The difference of simulated knockout and imputed counts (delta_X in CellOracle) was defined as velocity and the imputed counts from CellOracle were defined as the spliced moment Ms in scvelo. Transition probabilities were computed using scvelo's velocity_graph function with defaut parameters. Finally, the velocities are projected onto the scRNA atlas UMAP embedding using scvelo's velocity_embedding_stream function with lower velocities filtered out (min_mass = 3).

**Foxn1 knockout single-cell RNA data processing and quality control**. From the four libraries in the Foxn1 experiment, 29,276 cells were present in the filtered output from Cellranger. Preliminary clustering and visualization were done using a typical Seurat workflow similar to the scRNA atlas: cells were scaled to a total count of 10,000; then cells were natural log-transformed with a pseudo-count of one; 613 genes were selected (expression cutoff = 0.1; dispersion cutoff = 0.8); selected genes were used for scaled PCA; and 30 PC's were passed to UMAP and Louvain clustering (resolution = 1). Foxn1 was excluded from the selected genes. Prior to gene scaling, the total UMI count for each cell was regressed out.

Clusters were examined for the presence of low-quality cells and contaminants using a nearest-neighbor classifier trained on the unfiltered pharynx scRNA atlas, with binary training labels indicating which cells were retained. Six clusters (5206 cells) were removed due to low depth or presence of contaminants as follows: Cluster 0 (3362 cells), Cluster 10 (1150 cells), and Cluster 24 (318 cells) due to low depth; Cluster 27 (207 cells), Cluster 28 (132 cells), and Cluster 30 (37 cells) due to contamination (Supplementary Fig. 9).

In addition, any remaining cell with under 2000 UMIs was removed (2121 cells), and any remaining cell classified as a contaminant was removed (45 cells). This left 21,904 cells across 4 samples (E12.5het_replicate1 = 6805,

E12.5het_replicate2 = 2258, E12.5homo_replicate1 = 8070 and E12.5homo_replicate2 = 4771).

**Foxn1 knockout single-cell RNA data analysis**. The 21,904 filtered cells were classified according to pharynx atlas cluster labels and positioned on the pharynx atlas UMAP embedding. To do this, a 30-dimensional principal subspace was constructed by applying unscaled PCA to the variable genes already selected for the pharynx atlas. Foxn1 experiment cells were centered using the mean expression from the pharynx atlas, and they were projected into the pharynx atlas principal subspace using the projection matrix from the existing PCA. With both datasets represented with the same 30 latent components, 50 nearest neighbors in the atlas were calculated for each cell in the Foxn1 experiment. UMAP positions were computed by averaging across neighbors and labels were assigned by a plurality vote. Asymptotic 95% confidence intervals for cluster abundances were computed from a quasi-Poisson generalized linear model using sequencing date and perturbation as covariates. Total cell count by replicate was used as an offset. To propagate uncertainty in cluster assignments, total counts were based on probabilistic assignments, which were multiplied by the number of neighbors to yield whole numbers suitable for a Poisson response.

**Foxn1 knockout single-cell RNA differential expression testing**. For each cell, the "thymus probability score" (i.e., probability of being thymus) was computed as the sum of the probabilities for clusters 4, 25, and 9. 1173 cells were retained for differential expression analysis when two requirements were satisfied: thymus probability score at least 80% and detection of Il7 (> 0 counts) (Supplementary Fig. 9). To account for the substantial batch-level variation in the Foxn1 experiment, raw counts were summed within each sample, and differential expression between KO and Het was tested using edgeR[113] with sequencing date and genotype as covariates. Differentially expressed transcripts between Het and KO (log2-fold change > 0.5, FDR < 0.01) and between KO and Het (log2-fold change > 0.5, FDR < 0.01) were fed into Enrichr[114,115] via the enrichR R package (https://github.com/wjawaid/enrichR). Curated lists of the top 20 terms were visualized using custom dotplots.

To compare the *Foxn1* in silico with the experimental KO on the atlas scRNA variable genes in Fig. 5e, 991 genes were identified that were present in the atlas variable set as well as in the gene regulatory network and the differential test of the *Foxn1* KO single-cell data above.

The per gene simulated in silico perturbed expression values were averaged across all cells in thymic clusters 4, 9, and 25 and plotted on the X-axis while the corresponding fold-change value from the differential testing on the *Foxn1* KO single-cell data above were plotted on the Y axis. The Spearman rank correlation test was performed using the R function cor.test.

**Subset analysis of third pouches**. From the atlas overview, clusters 2, 4, 9, 25, and 26 were isolated, having in total 9256 cells. Scaled expression values from the overview were re-used (see the section "Atlas overview analysis").

Cell-by-gene count matrices of all samples were then concatenated to a single matrix. To account for differences in sequencing depth, UMI counts of each cell were normalized by total counts of that cell and values were log2-transformed. Highly variable genes ($n = 2139$) were selected using the same regression-based strategy as the atlas overview.

All further analyses were run with python3 using the scanpy package[106] (v1.4.6, https://github.com/theislab/scanpy) except stated otherwise. A single-cell neighborhood graph (kNN-graph) was computed on the 50 first principal components using 30 neighbors. For clustering, Louvain-based clustering[85], was used as implemented in louvain-igraph (v0.6.1 https://github.com/vtraag/louvain-igraph) and adopted by scanpy (tl.louvain). The resolution parameter was set to 1. For visualization, UMAP was run using the kNN-graph on the first 50 principal components. After Louvain clustering, predicted heterotypic doublets from doubletFinder (see the section "Single-cell RNA atlas quality control and filtering") and a small cluster enriched for them were removed leaving 8717 cells.

**Differential expression testing to describe populations in third pouches**. Differential expression testing between Louvain-based populations was performed on normalized and log2-transformed data using t-test with overestimated variance implemented in the tl.rank_genes_groups function of scanpy. P-values were corrected to control the false discovery rate using Benjamini–Hochberg method. The top 50 ranked genes per cluster are presented in Supplementary Dataset 9. To describe the progression from immature to mature cells and the associated gene expression changes, the trajectories were inferred using diffusion map from scanpy (tl.diffmap). We identified two trajectories in which cells transitioned from immature cells to thymus (T1) and parathyroid (T2). Cells were ordered based on a cell-to-cell distance metric using the concept of diffusion pseudotime (tl.dpt). We selected the cell with the highest Diffusion Component 2 (DC2) value (on y axis) within the starting population to act as root for the diffusion pseudotime.

**Integration of third pouches and Foxn1-deficiency experiment**. We integrated the third pouches (reference data) with the Foxn1 KO/Het data using Ingest from scanpy (tl.ingest) which allowed us to compute a UMAP embedding that included

both datasets. Ingest first projects Foxn1 KO/Het dataset on a PCA that has been fitted on the reference data using the highly variable genes of the reference data ($n = 2139$). It then uses a kNN classifier for mapping labels and the UMAP package for mapping the embeddings. The kNN classifier was computed on the first 50 principal components using 10 neighbors.

**Foxn1 knockout single-cell RNA staging relative to atlas**. For each genotype in the Foxn1 experiment, thymic cells were paired with equivalents in the atlas. Beginning with the pseudotime analysis of the third pouch (8717 cells), clusters 8, 9, and 2 of parathyroid and UBB-progenitor cells were removed. Remaining 7169 cells were grouped into 20 bins with an equal number of cells in each bin. Bulk profiles were computed by summing raw counts for each pseudotime bin and for each genotype in the Foxn1 experiment. A list of dynamic genes was selected and rank-transformed for each bulk profile, and Pearson correlations were computed between each genotype and each pseudotime bin. For each genotype, the two pseudotime bins with the largest correlations were selected as the atlas equivalent. Dynamic genes were identified using the thymusatlastools2 function GetDynamicGenes, with num_periods_initial_screen = 20. Genes were retained if max_fold_change > 0.75 and $q$-value < 0.01.

**Statistics and reproducibility**. No statistical methods were used to predetermine sample size. From each litter, we pooled the maximum number of embryos of the desired genotype ($Pax9^{VENUS/wt}$ or $Pax9^{VENUS/wt}Foxn1^{nu/(nu \text{ or } wt)}$). All samples consisted of pooled dissected pharyngeal endoderm tissue from multiple embryos, except for scRNA Foxn1 KO replicate 1, which was produced from a single embryo. Approximately 20,000 $Pax9^{VENUS}$ and Epcam-PE/Cy7 double-positive cells were collected per replicate, and all cells were used as input for 10X.

All attempts at replication were successful. Correlation across replicates of the same embryonic timepoint demonstrates reproducible data (Supplementary Fig. 1), and visualization of individual replicates by dataset shows strong overlap within timepoints/genotypes (Supplementary Figs. 2, 5, 10). All samples were collected on separate days, except for scRNA E10.5 replicates 1 and 2 and scATAC E12.5 replicates 1 and 2, which consisted of separate litters isolated on the same day, respectively. Samples of Foxn1 KO and heterozygous tissue were collected the same day, but replicates were performed on separate days. The single-cell RNA-seq pharyngeal endoderm dataset comprises two replicates at E9.5, two replicates at E10.5, three replicates at E11.5, and two replicates at E12.5. The single-cell ATAC-seq pharyngeal endoderm dataset comprises two replicates at E11.5 and two replicates at E12.5. The Foxn1 experiment single-cell RNA-seq dataset comprises two replicates of Foxn1 KO embryos and two replicates of Foxn1 heterozygous embryos, both at E12.5. For RNAscope experiments, two or more biological samples were stained.

The experiments were not randomized. All single-cell analyses were performed in a randomized manner for each of the three datasets, meaning cells from all samples were combined, analyzed, and allocated into clusters at a resolution which captured different cell types based on differentially expressed (or accessible in case of scATAC-seq) markers. Experimental group and sample information were controlled to overcome batch effects for the analysis of the scRNA datasets. No such batch correction was performed on the scATAC-seq dataset. For the Foxn1 knockout experiment, heterozygous littermates were used as controls.

For the scRNA and scATAC atlas datasets, blinding was not necessary as only one timepoint was collected on a given day. Since embryos of the same genotype were pooled in the Foxn1 scRNA experiments, genotyping was performed prior to tissue processing and thus the experiment could not be performed in a blinded manner. All embryos were processed by a single researcher. All scRNA and scATAC sequencing libraries were prepared and sequenced by a single researcher without blinding. Cell identity, embryonic day, and genotype were blinded during the scRNA and scATAC-seq analysis. For the scRNA-seq datasets, investigators were not blinded to experimental groups after processing as this would preclude grouping of replicates for analysis. For the scATAC-seq dataset, investigators were blinded to experimental groups even during the analysis as there was no need to correct for batch effects. Blinding was not required for the in situ hybridization experiments given that all stains were performed on wild-type samples. Blinding was not possible for the FACS and flow cytometry experiments because the genotype of each animal was determined prior to tissue processing. Genotyping and tissue processing were performed by a single researcher.

For data exclusions, please see the following sections: "Single-cell RNA atlas quality control and filtering", "Single-cell ATAC atlas data processing and quality control", "Foxn1 knockout single-cell RNA data processing and quality control", and "Subset analysis of third pouches".

**Reporting summary**. Further information on research design is available in the Nature Research Reporting Summary linked to this article.

## Data availability

The scRNA-seq and scATAC-seq short reads and count matrices data generated in this study have been deposited in the Gene Expression Omnibus (GEO) database under accession codes "GSE182135" (scRNA atlas), "GSE182136" (scRNA knockout experiment), and "GSE182134" (scATAC atlas). The motif data used in the scATAC analysis is available

in the Cis-BP version 2.00 database for "Mus musculus [http://cisbp.ccbr.utoronto.ca/bulk.php]". The conservation scores used in the scATAC analysis are available in the PhastCon score database for the "Euarchontoglires clade [http://hgdownload.cse.ucsc.edu/goldenpath/mm10/phastCons60way/mm10.60way.phastCons60wayEuarchontoGlire.bw]". All other relevant data supporting the key findings of this study are available within the article and its Supplementary Information files or from the corresponding author upon reasonable request.

## Code availability

The code (custom scripts and packages) and documentation are available at the GitHub[116,117] repositories [https://github.com/maehrlab/pharyngeal_endoderm_development] and [https://github.com/maehrlab/thymusatlastools2].

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

## Acknowledgements

This work was supported by R01AI132963, U01DK104218, a grant from the U.S.-Israel Binational Science Foundation, and a research grant from the University of Pennsylvania Orphan Disease Center in partnership with the Hypopara Research Foundation to R.M., by T32 AI132152 pre-doctoral training support to M.E.M., and by Helmholtz Association's Initiative and Networking Fund through Helmholtz AI [grant number: ZT-I-PF-5-01] and sparse2big [grant number ZT-I-007] to F.J.T. We are grateful to P. Zamore and J. Luban for sequencer access, the UMass Medical School FACS Core for support with cell sorting, and the UMass Medical School Histology Core for support with tissue sections.

## Author contributions

M.E.M., M.L., E.M.K., H.A., F.J.T., and R.M. wrote and edited the manuscript. The experiments were designed, analyzed, and interpreted by M.E.M., M.L., E.M.K., and R.M. M.E.M. and T.J.P. performed the experiments. M.L., E.M.K., H.A., and J.H. performed bioinformatic data processing and analyses.

## Competing interests

F.J.T. reports receiving consulting fees from Immunai and ownership interest in Dermagnostix GmbH and Cellarity. All other authors declare no competing interests.

## Additional information

**Peer review information** *Nature Communications* thanks Esther Wershoff and the other anonymous reviewer(s) for their contribution to the peer review this work. Peer reviewer reports are available.

