## [Peer Review File · Nature Communications]

Reviewers' Comments:

Reviewer #1:

Remarks to the Author:

This manuscript reports the collection, validation and initial analysis of an important new resource - a single cell RNA seq and ATACseq dataset for the developing pharyngeal endoderm, covering day 9.5 to 12.5 of mouse embryonic development and including comparative analysis of pharyngeal endoderm development in wild type and *Foxn1* null mice across part of this time period (E11.5 and E12.5). The analysis of these datasets presented identifies and validates (by RNAscope) clusters corresponding to the different organ primordia and identifies from the dataset novel markers for some of the clusters, including a putative new transcriptional regulator of the medullary thymic epithelial sub-lineage. It also identifies promoters and distal regulatory elements associated with genes expressed in each cluster, including some putative cell type-specific regulatory elements. Further analysis begins to identify GRNs for some of the organ primordia (thymus, parathyroid and thyroid). The analysis is taken further for thymic primordium via addition of *Foxn1*^{+/-} and *Foxn1*^{-/-} scRNAseq data at E11.5 and E12.5. Analysis of these data supports the existing model in which the absence of *Foxn1* imposes a developmental block (or strong delay) on thymic epithelial lineage development, and provides support and insight into the recently presented hypothesis that *FOXN1* expression in the thymic epithelial lineage results in suppression of a branching morphogenesis programme. It also identifies *FOXN1* as a regulator of genes associated with EMT, including *Snai2* and *Zeb2*. The study further tests and validates a modelling approach in which the effect of lack of *Foxn1* is simulated using GRN predictions, indicating the potential of this type of approach for further analysis of this and other datasets.

Overall, the paper is well written, the data presented support the conclusions drawn (with a minor caveat) and the work appears well done. The datasets themselves will be an important resource for the field, with possible relevance to understanding human developmental disorders as well as the organogenesis of the organs deriving from this region (thymus, parathyroid, thyroid, ultimobranchial bodies, pharynx). Additionally, the analyses presented already point to previously uncharacterised facets of thymus organogenesis.

Minor comments

Fig 1: Please state somewhere in the text how many cells from each timepoint were represented in the scRNAseq analysis (after QC and removal of contaminants). A figure showing the projections of the reps with more clarity would also be welcome (eg a supplementary fig showing each rep projected individually on to the total).

Additionally, since the lineage relationships in the E12.5 thymus are not yet dissected in any detail, despite the clustering data, the description of *Grhl3* as a novel mTEC lineage specific TF seems premature and should be toned down slightly (though it may well turn out to be true); related, although it is difficult to tell from a single section, *Grhl3* seems more broadly expressed in the RNAscope image than one would expect for an mTEC-lineage restricted marker.

Fig 2: Similarly, please state somewhere how many cells were in each of the identified organ clusters at each age.

Fig 5b: From the data, the *Foxn1* het appears to be missing mTEC and most cTEC, although it is more developmentally advanced than the KO. This expected but is not mentioned in the text.

Reviewer #2:

Remarks to the Author:

Review manuscript NCOMMS-21-32924-T

In this study the authors perform a comprehensive single cell transcriptomic and chromatin accessibility analysis of the developing pharyngeal endoderm. The primary strength of this manuscript is the generation of single cell datasets that would be useful to the community in

future investigations of endoderm development. The authors also identify a new lineage specific marker *grhl3* and cluster specific distal regulatory elements. However, as these findings are based primarily on in-silico analysis, their utility will depend on future experimental validation of the identified regulatory elements.

Specific comments:

- Given the large number of specialized tools and programs used in the manuscript it is important to briefly explain some of the approaches and technical terms in the main text. While the methods section describes the computational approaches in detail, the manuscript is currently hard to follow for a reader not familiar with the field and some of the changes mentioned below can improve readability.
- For scRNA-Seq it is important to clarify how the differential analysis was performed. The genes shown in Figure 1D were shortlisted based on a fold change and p-value cut-offs. How was this differential analysis conducted? Is it based on pairwise differential comparisons between the clusters, comparison against a background set or some other approach? Similarly, Figure 2F shows shortlisted genes based on scATAC-Seq differential analysis which should also be defined more clearly.
- In figure 1C please include specific gene markers used for classification along with cluster labels
- Line 97. The peripheral cluster being referred to should be labeled either in supplementary or main figure
- Briefly describe how the cluster specificity index in figure 3B was calculated
- The 'Atlas qc and filtering' section in the methods mentions that several clusters with high or low expression of specific genes (such as *Neurod1* and *Pax9*) were discarded. Can the authors comment on the rationale for this approach?
- Since *Grhl3* is proposed to be a novel marker for mTEC cells, the corresponding UMAP and gene activity/motif enrichments visualizations for this gene should be included in Figure 1 and Figure 2 g,h,i
- It is not clear what the dot sizes based on the correlation values in the legend of Figure 3C represent. Why are all the correlation values negative and greater than 1?

Reviewer #3:

Remarks to the Author:

In "Integrated single-cell transcriptomic and chromatin accessibility approaches reveal regulatory programs driving pharyngeal organ development" the authors use scRNA-seq and scATAC-seq data to investigate pharyngeal endodermal development between E9.5-E12.5. The transcriptomic data reveals novel cell type-specific gene expression patterns and they authors combine the -omic data types to reveal gene regulatory networks (GRN) governing development in the pharyngeal region. Finally, they used their GRN predictions on transcription factor *Foxn1* to unpick deviations from the normal genetic program in *Foxn1* KO mice.

Overall, this is a thorough study addressing an important question in the developmental field and can act as a framework for similar analysis that ought to be done on other endodermal organs. The analysis is rigorous, and conclusions drawn from it are logical. The manuscript was enjoyable to read and this paper makes a significant contribution to the field.

Major points

Line 106/107 "proximity on the UMAP reflects co-development of these organs within the third pharyngeal pouch". *UBB* is also close on the UMAP. Can the authors use Palantir or a similar

method to map development trajectory towards MTECs/CTECs?

Why was it necessary to use two methods for GRN inference? Can the author derive the same results just with CellOracle? If not, need to explain fully why both are complementary. How exactly does the first method deviate from SCENIC?

Line 221 – “Foxn1 KO mice fail to develop a functional thymus”. The subsequent analysis on Foxn1 is nice but not novel, but are there any other TFs that the GRN predicts would prevent development of the thymus? Could the authors try any of these knockouts or at least discuss them as future avenues of exploration?

Minor points

Add a short description of pharyngeal endoderm development in the introduction for non-experts. What time points is this happening (you allude to this in line 79 but it should come earlier)? What is the rough morphology e.g. explain the pouches (how many, when, purpose etc.). Fig 1C schematic should be annotated better (e.g. with pouch number) and indicating what day this corresponds to.

Line 101 – “lacks Hoxa3 suggesting an anterior position” – explain why

Line 117 – “with known tissue-specific transcripts...” say what these genes are and are supposed to show (do this in the Figure legend too). In Fig 1D what is the Foxn1 showing/ why do you show it here? Why is Foxn1 expression surrounding Grh13 expression? Interpretation?

Line 129 – why didn't the authors get earlier timepoints for scATAC-seq? Wouldn't you expect to see important regulatory mechanisms at work at E9.5 and E10.5? Need to justify this.

Line 136 – first time using TSS acronym, introduce it.

Line 139 – Fig 2f out of order (2d and 2e come after)

Line 248 – Fig 5e is not completely convincing. What is the p-value? Why so noisy? Which genes are predicted better than others, and do you have any sense of why this is?

Line 252 – typo with references

Line 280 – Hippo discovery seems important, can you elaborate/suggest future directions with this in discussion?

Supplementary Fig 4 – hard to read, low quality, remake

Online methods

line 123-130 – explain why presence/lack of those genes led you to exclude those clusters.

Line 139 – why did you use regression-based batch correction and how does it compare to another method such as MNN? Please provide a comparison

Line 151 - instead of variance, used variance of residuals. Explain why.

Line 314 – State explicitly if/how this method deviates from SCENIC

Line 393 – how robust is a. choice of 50 nearest neighbors?

Line 404 - refer to relevant figure

Line 441 – What is DC1?

Response to the reviewers:

We thank the reviewers for the constructive and insightful feedback, which has helped us strengthen the manuscript. We are pleased to have received an overall positive response to our study and have addressed the remaining specific comments in a point-by-point manner below.

REVIEWER COMMENTS

Reviewer #1 (Remarks to the Author):

This manuscript reports the collection, validation and initial analysis of an important new resource - a single cell RNA seq and ATACseq dataset for the developing pharyngeal endoderm, covering day 9.5 to 12.5 of mouse embryonic development and including comparative analysis of pharyngeal endoderm development in wild type and *Foxn1* null mice across part of this time period (E11.5 and E12.5). The analysis of these datasets presented identifies and validates (by RNAscope) clusters corresponding to the different organ primordia and identifies from the dataset novel markers for some of the clusters, including a putative new transcriptional regulator of the medullary thymic epithelial sub-lineage. It also identifies promoters and distal regulatory elements associated with genes expressed in each cluster, including some putative cell type-specific regulatory elements. Further analysis begins to identify GRNs for some of the organ primordia (thymus, parathyroid and thyroid). The analysis is taken further for thymic primordium via addition of *Foxn1*^{+/-} and *Foxn1*^{-/-} scRNAseq data at E11.5 and E12.5. Analysis of these data supports the existing model in which the absence of *Foxn1* imposes a developmental block (or strong delay) on thymic epithelial lineage development, and provides support and insight into the recently presented hypothesis that *FOXN1* expression in the thymic epithelial lineage results in suppression of a branching morphogenesis programme. It also identifies *FOXN1* as a regulator of genes associated with EMT, including *Snai2* and *Zeb2*. The study further tests and validates a modelling approach in which the effect of lack of *Foxn1* is simulated using GRN predictions, indicating the potential of this type of approach for further analysis of this and other datasets.

Overall, the paper is well written, the data presented support the conclusions drawn (with a minor caveat) and the work appears well done. The datasets themselves will be an important resource for the field, with possible relevance to understanding human developmental disorders as well as the organogenesis of the organs deriving from this region (thymus, parathyroid, thyroid, ultimobranchial bodies, pharynx). Additionally, the analyses presented already point to previously uncharacterised facets of thymus organogenesis.

Minor comments

Fig 1: Please state somewhere in the text how many cells from each timepoint were represented in the scRNAseq analysis (after QC and removal of contaminants). A figure showing the projections of the reps with more clarity would also be welcome (eg a supplementary fig showing each rep projected individually on to the total).

Response: We agree with the reviewer that these suggested changes will make the resource more accessible. Therefore, we have amended the main text, online methods, and figures to address these suggestions.

For the single-cell transcriptomic atlas, we have edited the main text to include the number of cells at each timepoint for Fig. 1. We have also edited the “Single-cell RNA atlas quality control and filtering” methods section to include the number of cells from each replicate.

Similarly, for the scATAC dataset in Fig. 2, we have edited the main text to include the number of cells at each timepoint. We have also edited the “Single-cell ATAC atlas data processing and quality control” methods section to include the number of cells from each replicate.

For the *Foxn1* knockout single-cell RNA experiment, we have edited the main text to state the number of cells for each condition. We have also edited the “*Foxn1* knockout single-cell RNA data processing and quality control” methods section to include the number of cells from each replicate.

All the above-mentioned changes are highlighted in yellow throughout the main text and online methods.

Furthermore, we have created three new supplementary figures to highlight each replicate projected onto the full UMAP for every dataset. The single-cell RNA atlas replicates are included in Supplementary Fig. 2, the single-cell ATAC replicates are included in Supplementary Figure 5, and the *Foxn1* experiment single-cell RNA replicates are included in Supplementary Fig. 10.

Additionally, since the lineage relationships in the E12.5 thymus are not yet dissected in any detail, despite the clustering data, the description of *Grhl3* as a novel mTEC lineage specific TF seems premature and should be toned down slightly (though it may well turn out to be true); related, although it is difficult to tell from a single section, *Grhl3* seems more broadly expressed in the RNAscope image than one would expect for an mTEC-lineage restricted marker.

Response: We appreciate the reviewer’s input on this matter and have altered the sentence to reflect a more conservative conclusion from the RNAscope data: “While *Gas6* expression overlaps with *Foxn1* throughout the thymus organ domain, we find that mTECs mainly express *Grhl3*, which could be a novel lineage-specific TF (Fig. 1f).”

Fig 2: Similarly, please state somewhere how many cells were in each of the identified organ clusters at each age.

Response: To better describe cluster composition by embryonic day, we have produced a new supplementary table (Supplementary Table 1) detailing the number of cells by embryonic day per cluster. Clusters with identifiable cell types have been labeled according with the cluster labels in Fig. 1c.

Fig 5b: From the data, the *Foxn1* het appears to be missing mTEC and most cTEC, although it is more developmentally advanced than the KO. This expected but is not mentioned in the text.

Response: To highlight this observation, we have amended the sentence and included an additional citation to support the finding. The sentence now reads: “Notably, while the *Foxn1* heterozygous control sample appeared more mature than the *Foxn1* KO, the control included less mature TECs as compared to the atlas, a result that agrees with previous descriptions of reduced cellularity in the *Foxn1* heterozygous thymus⁶⁰.”

Reviewer #2 (Remarks to the Author):

In this study the authors perform a comprehensive single cell transcriptomic and chromatin accessibility analysis of the developing pharyngeal endoderm. The primary strength of this manuscript is the generation of single cell datasets that would be useful to the community in future investigations of endoderm development. The authors also identify a new lineage specific marker *grhl3* and cluster specific distal regulatory elements. However, as these findings are based primarily on in-silico analysis, their utility will depend on future experimental validation of the identified regulatory elements.

Specific comments:

- Given the large number of specialized tools and programs used in the manuscript it is important to briefly explain some of the approaches and technical terms in the main text. While the methods section describes the computational approaches in detail, the manuscript is currently hard to follow for a reader not familiar with the field and some of the changes mentioned below can improve readability.

Response: We thank the reviewer for their feedback and have updated the main text to include brief explanations of methodologies used. We had to balance manuscript length with depth of added explanations and hope the updated manuscript will now be more accessible.

- For scRNA-Seq it is important to clarify how the differential analysis was performed. The genes shown in Figure 1D were shortlisted based on a fold change and p-value cut-offs. How was this differential analysis conducted? Is it based on pairwise differential comparisons between the clusters, comparison against a background set or some other approach? Similarly, Figure 2F shows shortlisted genes based on scATAC-Seq differential analysis which should also be defined more clearly.

Response: We thank the reviewer for pointing out the need for a detailed description of the differential gene expression analyses. We have updated the “Single-cell RNA atlas overview analysis” methods section with differential expression analysis details for Fig. 1d, Supplementary Fig. 3, Supplementary Table 2. We have also updated the “Single-cell ATAC atlas analysis” methods section with differential expression analysis Fig. 2f and Supplementary Table 3. The additions are highlighted in yellow.

- In figure 1C please include specific gene markers used for classification along with cluster labels

Response: To emphasize genes that were used for cluster annotation, we edited Fig. 1e to include red asterisks next to cluster-specific markers with documentation in the literature. Additionally, in Supplementary Fig. 3 we extended the dotplot to show the expression pattern of each gene discussed in the main text as well as additional relevant genes described in the literature.

- Line 97. The peripheral cluster being referred to should be labeled either in supplementary or main figure

Response: We thank the reviewer for helping us improve the readability of this manuscript. To clarify the reference to a “peripheral cluster”, we have amended the text to link the sentence to the existing label on the figure. This line now reads: “The second pouch cluster comprising mainly E10.5 and E11.5 cells expresses genes associated with salivary glandular epithelium (*Irf6*²⁸, *Trp63*, *Sox9*, *Vim*²⁹).”

- Briefly describe how the cluster specificity index in figure 3B was calculated

Response: To improve the readability of our manuscript, we added a line in the main text explaining how this was calculated. The cluster specificity index is based on the tissue specificity index method described previously¹. We now have added clarifications regarding this index in the main text, and another description can also be found in the “Single-cell ATAC atlas analysis” online methods section.

- The ‘Atlas qc and filtering’ section in the methods mentions that several clusters with high or low expression of specific genes (such as *Neurod1* and *Pax9*) were discarded. Can the authors comment on the rationale for this approach?

Response: A drawback of FACS enrichment is that the sorted population may contain contaminant cells besides the pharyngeal endoderm. Thus, during the data cleaning process we removed such contaminants. Of note, count matrices for all cells have been deposited at GEO, so future analyses may choose to include such cells.

To clarify our rationale for labeling certain clusters as contaminants and subsequently removing such clusters during data cleaning, we have added citations in the online methods section to provide support for our conclusions. These changes can be found highlighted in yellow in the “Single-cell RNA atlas quality control and filtering” online methods section.

- Since *Grhl3* is proposed to be a novel marker for mTEC cells, the corresponding UMAP and gene activity/motif enrichments visualizations for this gene should be included in Figure 1 and Figure 2 g,h,i

Response: We updated Supplementary Fig. 3 to include UMAPs of each novel marker that we verified with RNAscope, including *Grhl3*. At this point, we did not update Fig. 2 g-i with gene activity or motif enrichment plots of *Grhl3*, because the Cis-BP motif database that we utilized lacks a *Grhl3* motif. We think this is acceptable since investigation of *Grhl3* regulatory activity is not within the scope of this resource. As such, we have softened the wording around *Grhl3*-related mechanisms.

- It is not clear what the dot sizes based on the correlation values in the legend of Figure 3C represent. Why are all the correlation values negative and greater than 1?

Response: We thank the reviewer for their feedback and agree that the figure lacked clarity. The correlation values were plotted on a 0-100 scale and appeared negative due to formatting issues. To resolve these issues and improve the clarity of this figure, we have presented the results as a heatmap with a colorbar, rather than a dotplot.

In the updated Fig. 3c panel, the heat represents the scaled motif score (which was originally the color of the dot) and the correlation scale represents the correlation between the motif score and the integrated gene expression for the maximally correlated pair.

Reviewer #3 (Remarks to the Author):

In “Integrated single-cell transcriptomic and chromatin accessibility approaches reveal regulatory programs driving pharyngeal organ development” the authors use scRNA-seq and scATAC-seq data to investigate pharyngeal endodermal development between E9.5-E12.5. The transcriptomic data reveals novel cell type-specific gene expression patterns and they authors combine the -omic data types to reveal gene regulatory networks (GRN) governing development in the pharyngeal region. Finally, they used their GRN predictions on transcription factor *Foxn1* to unpick deviations from the normal genetic program in *Foxn1* KO mice.

Overall, this is a thorough study addressing an important question in the developmental field and can act as a framework for similar analysis that ought to be done on other endodermal organs. The analysis is rigorous, and conclusions drawn from it are logical. The manuscript was enjoyable to read and this paper makes a significant contribution to the field.

Major points

Line 106/107 “proximity on the UMAP reflects co-development of these organs within the third pharyngeal pouch”. UBB is also close on the UMAP. Can the authors use Palantir or a similar method to map development trajectory towards MTECs/CTECs?

Response: The reviewer refers to a statement that is inaccurate, and we thank the reviewer for calling attention to this. The specific distance between two cells on a 2D UMAP embedding is not synonymous with co-development or proximity within a tissue; therefore, we have revised the sentence by removing the statement in question:

“In agreement with previous reports^{37,38}, parathyroid (*Gcm2*) and thymus (*Foxn1*) appear by E10.5 and E11.5, respectively (Fig. 1), and proximity on the UMAP reflects co-development of these organs within the third pharyngeal pouch²⁷.”

Pseudotime analysis between the thymus and parathyroid is included in Supplementary Fig. 11.

Why was it necessary to use two methods for GRN inference? Can the author derive the same results just with CellOracle? If not, need to explain fully why both are complementary. How exactly does the first method deviate from SCENIC?

Response: Our approach of using two methods for GRN inference was necessary to emphasize both commonalities as well as differences across clusters captured in our data. We first implemented GENIE3 to derive a global GRN framework of pharyngeal endoderm development. In the main figure, we highlight certain cell type-specific subnetworks captured by GENIE3, but this approach also identified subnetworks that were shared across cell types. To better emphasize this result, we added a supplementary figure (Supplementary Fig. 6) that includes UMAPs displaying expression patterns of each subnetwork derived from GENIE3. These results allowed us to better understand gene signatures shared across clusters, thereby improving our understanding of how each cluster relates to others. While GENIE3 did highlight certain cell type-specific networks as described in the main figure, the GENIE3 network lacked the resolution to examine all unique cell states captured in our data. Furthermore, the GENIE3 results could be difficult to interpret when one TF performs multiple roles in different cell types. For instance, *Nkx2.1* regulates both UBB and thyroid development, and thus will have connections relevant to each cell type that could be indistinguishable. Finally, the GENIE3 GRN is built solely on transcriptomic data. Thus, we also implemented CellOracle, which is constrained by pre-defined clusters, to expose all cell type-specific GRNs at play during pharyngeal endoderm differentiation. CellOracle provides the added benefits of 1) integrating chromatin accessibility information, which allowed us to leverage our pharyngeal endoderm single cell chromatin data, and 2) simulating genetic knockouts.

A description of the contrast between both approaches can be found in the main text.

Line 221 – “*Foxn1* KO mice fail to develop a functional thymus”. The subsequent analysis on *Foxn1* is nice but not novel, but are there any other TFs that the GRN predicts would prevent development of the thymus? Could the authors try any of these knockouts or at least discuss them as future avenues of exploration?

Response: While it is beyond the scope of this resource to mechanistically characterize previously unknown factors involved in thymus organogenesis, we now have added extra simulations of TF knockouts predicted from our GRNs for several organ domains, including the thymus, parathyroid, and thyroid. These can be found in Supplementary Fig. 7. At the suggestion of the reviewer, we have also amended the discussion to consider future applications and avenues of knockout simulations.

Minor points

Add a short description of pharyngeal endoderm development in the introduction for non-experts. What

time points is this happening (you allude to this in line 79 but it should come earlier)? What is the rough morphology e.g. explain the pouches (how many, when, purpose etc.). Fig 1C schematic should be annotated better (e.g. with pouch number) and indicating what day this corresponds to.

Response: We thank the reviewer for this comment aimed at helping us improve the readability of our manuscript. We have added a short section at the beginning of the manuscript to address this comment:

“Development of the pharyngeal endoderm, which produces specialized epithelia of various organs including the thymus, parathyroid, thyroid, ultimobranchial body, the middle ear, and palatine tonsils¹⁻³ is characterized by dramatic changes in morphology and cell state across embryonic days (E)9.5 to 12.5 of mouse gestation. At E9.5, pharyngeal endoderm comprises four bilateral out-pockets organized from anterior to posterior, termed pharyngeal pouches I – IV. Each pouch forms the epithelial structure of a unique organ that separates from the pharynx by E12.5.”

We have also improved the labeling of Fig. 1d (originally Fig. 1c) to highlight the pouches as well as the endoderm tissue relative to the mesoderm and ectoderm tissues.

Line 101 – “lacks *Hoxa3* suggesting an anterior position” – explain why

Response: We thank the reviewer for helping us clarify this conclusion. Previous work in the field, including *in situ* hybridization experiments as well as various genetic knockout experiments, demonstrates that *Hoxa3* is expressed in the third and fourth pharyngeal pouches, but not the first or second pouches, and is important for proper thymus and parathyroid development²⁻⁴. The sentence now reads: “This cluster also expresses pharyngeal pouch markers *Pax1* and *Fgf8*^{31,32}, but lacks expression of *Hoxa3*, a known marker of the caudal third and fourth pouches³³, suggesting an anterior position.”

Line 117 – “with known tissue-specific transcripts...” say what these genes are and are supposed to show (do this in the Figure legend too). In Fig 1D what is the *Foxn1* showing/ why do you show it here? Why is *Foxn1* expression surrounding *Grh13* expression? Interpretation?

Response: To clarify the reference to “known tissue-specific transcripts”, we have amended the main text as follows:

“To verify these findings, we implemented RNAscope at E12.5. Using *Foxn1*, *Gcm2* and *Calca* probes to demarcate the thymus, parathyroid and UBB, respectively, we co-stained with predicted transcripts enriched in the thymus (*Gas6*, *Grhl3*), parathyroid (*Sparcl1*, *Ibsp*, *Flrt2*) and UBB (*Meox1*) (Fig. 1 f-h, Supplementary Fig. 2). While *Gas6* expression overlaps with *Foxn1* throughout the thymus organ domain, we find that *Grhl3* expression is largely restricted to the thymic medulla, which together with the single cell transcriptomic analysis suggests mTEC specificity (Fig. 1f).”

Additionally, we have updated the corresponding figure legend with the names of each known transcript.

Line 129 – why didn't the authors get earlier timepoints for scATAC-seq? Wouldn't you expect to see important regulatory mechanisms at work at E9.5 and E10.5? Need to justify this.

Response: Based on our analysis of the single cell transcriptomic data, we identified organ-specific gene signatures most prevalently between days E11.5 and E12.5. As such, we focused on those two timepoints and produced in-depth single cell chromatin accessibility data. The rationale for this decision is also mentioned in the main text.

Line 136 – first time using TSS acronym, introduce it.

Response: We thank the reviewer for their attention to detail and have now introduced the acronym when it is first mentioned.

Line 139 – Fig 2f out of order (2d and 2e come after)

Response: We thank the reviewer for their attention to detail. The main text has been amended to have sentences referring to Fig. 2d-f in the correct order.

Line 248 – Fig 5e is not completely convincing. What is the p-value? Why so noisy? Which genes are predicted better than others, and do you have any sense of why this is?

Response: We have added an explanation, which is highlighted in yellow, in the “Foxn1 knockout single-cell RNA differential expression testing” online methods section explaining how we arrived at this figure. We have also added the p-value of the spearman correlation test ($p < 2.2e^{-16}$) to Fig. 5e.

With regards to noise and predictability, GRN models have several limitations. For example, the CellOracle GRN uses a linear regression model. Thus, it cannot model non-linear or combinatorial binding effects, as noted by the developers of CellOracle⁵, which we cited in our manuscript. It is also discussed by the CellOracle developers that an *in silico* perturbation simulation can only predict results where the perturbed state has been sampled in the original data. While we feel that the discussion of the general state of the GRN modeling field is beyond the scope of this publication, we have amended the discussion to include some of the caveats.

Line 252 – typo with references

Response: We thank the reviewer for their attention to detail and have corrected the error.

Line 280 – Hippo discovery seems important, can you elaborate/suggest future directions with this in discussion?

Response: We agree that this finding is exciting and could be important. To emphasize this finding further, we have included a new sentence in the discussion to highlight a potential role of Hippo signaling during thymus development: “Interestingly, Hippo signaling was active in the developing thymus and lacking in the *Foxn1* KO thymus, indicating that this pathway could be functioning downstream of *Foxn1* to regulate thymic proliferation and organ growth.”

Supplementary Fig 4 – hard to read, low quality, remake

Response: We have remade this figure, which has been renumbered as Supplementary Fig. 8, to improve the quality.

Online methods

line 123-130 – explain why presence/lack of those genes led you to exclude those clusters.

Response: A drawback of FACS enrichment is that the sorted population may contain contaminant cells besides the pharyngeal endoderm. Thus, during the data cleaning process we removed such contaminants. Of note, count matrices for all cells have been deposited at GEO, so future analyses may choose to include such cells.

To clarify our rationale for labeling certain clusters as contaminants and subsequently removing such clusters during data cleaning, we have added citations in the online methods section to provide support for our conclusions. These changes can be found highlighted in yellow in the “Single-cell RNA atlas quality control and filtering” online methods section.

Line 139 – why did you use regression-based batch correction and how does it compare to another method such as MNN? Please provide a comparison

Response: Other work by Luecken *et al* and Buttner *et al*^{6,7} has compared regression-based batch correction with non-linear methods such as MNN, which revealed that regression-based approaches perform reasonably well and sometimes even outperform non-linear approaches such as MNN. We have successfully implemented regression-based correction in Kernfeld *et al*⁸ and found it also useful to simultaneously address confounding effects such as sequencing depth and cell cycle. We have deposited the raw data to GEO if others would prefer alternative batch correction methods.

Line 151 - instead of variance, used variance of residuals. Explain why.

Response: We want to account for corrected expression values (after correction for batch effects and nuisance variables) when selecting highly variable gene features for downstream analysis. Hence, we used the variance of the residuals after regression-based correction instead of the variance before correction.

Line 314 – State explicitly if/how this method deviates from SCENIC

Response: Our approach of using two methods for GRN inference was necessary to emphasize both commonalities as well as differences across clusters captured in our data. We first implemented GENIE3 to derive a global GRN framework of pharyngeal endoderm development. In the main figure, we highlight certain cell type-specific subnetworks captured by GENIE3, but this approach also identified subnetworks that were shared across cell types. To better emphasize this result, we added a supplementary figure (Supplementary Fig. 6) that includes UMAPs displaying expression patterns of each subnetwork derived from GENIE3. These results allowed us to better understand gene signature shared across clusters, thereby improving our understanding of how each cluster relates to others. While GENIE3 did highlight certain cell type-specific networks as described in the main figure, the GENIE3 network lacked the resolution to examine all unique cell states captured in our data. Furthermore, the GENIE3 results could be difficult to interpret when one TF performs multiple roles in different cell types. For instance, Nkx2.1 regulates both UBB and thyroid development, and thus will have connections relevant to each cell type that could be indistinguishable. Finally, the GENIE3 GRN is built solely on transcriptomic data. Thus, we also implemented CellOracle, which is constrained by pre-defined clusters, to expose all cell type-specific GRNs at play during pharyngeal endoderm differentiation. CellOracle provides the added benefits of 1) integrating chromatin accessibility information, which allowed us to leverage our pharyngeal endoderm single cell chromatin data, and 2) simulating genetic knockouts.

A description of the contrast between both approaches can be found in the main text.

Line 393 – how robust is a. choice of 50 nearest neighbors?

Response: As outlined in Luecken *et al*⁷, a reasonable selection for nearest neighbors is between 5 and 100 depending on the size of the dataset. Since we have a large dataset of 54,044 cells in the single cell transcriptomic atlas, we selected approximately 0.1% of the number of cells in the atlas i.e. 50 nearest neighbors. We do not believe this choice will alter the biological conclusions, since, for a more detailed investigation and staging of the knockout data, we further analyzed a subset of the cells in Fig. 6 and Supplementary Fig. 11. Count matrices for all cells have been deposited at GEO, so future analyses may choose to adjust this parameter.

Line 404 - refer to relevant figure

Response: We have adjusted the “Foxn1 knockout single-cell RNA differential expression testing” online methods section to include a reference to Supplementary Fig. 9, and we thank the reviewer for helping

us improve the readability of our manuscript.

Line 441 – What is DC1?

Response: We have edited the figure legend for Supplementary Fig. 11 to include the Diffusion component (DC) axes and have also edited the “Differential expression testing to describe populations in third pouches” online methods section to illustrate this.

Sources

- 1 Yanai, I. *et al.* Genome-wide midrange transcription profiles reveal expression level relationships in human tissue specification. *Bioinformatics* **21**, 650-659, doi:10.1093/bioinformatics/bti042 (2005).
- 2 Chisaka, O. & Capecchi, M. R. Regionally restricted developmental defects resulting from targeted disruption of the mouse homeobox gene *hox-1.5*. *Nature* **350**, 473-479, doi:10.1038/350473a0 (1991).
- 3 Manley, N. R. & Capecchi, M. R. The role of *Hoxa-3* in mouse thymus and thyroid development. *Development (Cambridge, England)* **121**, 1989-2003 (1995).
- 4 Chojnowski, J. L. *et al.* Multiple roles for *HOXA3* in regulating thymus and parathyroid differentiation and morphogenesis in mouse. *Development (Cambridge, England)* **141**, 3697-3708, doi:10.1242/dev.110833 (2014).
- 5 Kamimoto, K., Hoffmann, C. M. & Morris, S. A. CellOracle: Dissecting cell identity via network inference and in silico gene perturbation. *bioRxiv*, 2020.2002.2017.947416, doi:10.1101/2020.02.17.947416 (2020).
- 6 Buttner, M., Miao, Z., Wolf, F. A., Teichmann, S. A. & Theis, F. J. A test metric for assessing single-cell RNA-seq batch correction. *Nature methods* **16**, 43-49, doi:10.1038/s41592-018-0254-1 (2019).
- 7 Luecken, M. D. & Theis, F. J. Current best practices in single-cell RNA-seq analysis: a tutorial. *Mol Syst Biol* **15**, e8746, doi:10.15252/msb.20188746 (2019).
- 8 Kernfeld, E. M. *et al.* A Single-Cell Transcriptomic Atlas of Thymus Organogenesis Resolves Cell Types and Developmental Maturation. *Immunity* **48**, 1258-1270.e1256, doi:10.1016/j.immuni.2018.04.015 (2018).

Reviewers' Comments:

Reviewer #1:

Remarks to the Author:

The revised manuscript addresses my concerns.

Reviewer #2:

Remarks to the Author:

In the revised manuscript the authors have thoroughly addressed my initial concerns

Reviewer #3:

None